# CLASS INCREMENTAL LEARNING VIA LIKELIHOOD RATIO BASED TASK PREDICTION

**Haowei Lin[1], Yijia Shao[2], Weinan Qian[3], Ningxin Pan[3], Yiduo Guo[3], and Bing Liu[4]***

[1]Institute for Artificial Intelligence, Peking University    [2]Stanford University
[3]Wangxuan Institute of Computer Technology, Peking University
[4]Department of Computer Science, University of Illinois at Chicago
[1]`linhaowei@pku.edu.cn`  [2]`shaoyj@stanford.edu`
[3]`{ypqwn, 2100017816, yiduo}@stu.pku.edu.cn`  [4]`liub@uic.edu`

## ABSTRACT

*Class incremental learning* (CIL) is a challenging setting of continual learning, which learns a series of tasks sequentially. Each task consists of a set of unique classes. The key feature of CIL is that no task identifier (or task-id) is provided at test time. Predicting the task-id for each test sample is a challenging problem. An emerging theory-guided approach (called TIL+OOD) is to train a task-specific model for each task in a shared network for all tasks based on a *task-incremental learning* (TIL) method to deal with *catastrophic forgetting*. The model for each task is an *out-of-distribution* (OOD) detector rather than a conventional classifier. The OOD detector can perform both *within-task* (*in-distribution* (IND)) class prediction and OOD detection. The OOD detection capability is the key to task-id prediction during inference. However, this paper argues that using a traditional OOD detector for task-id prediction is sub-optimal because additional information (e.g., the replay data and the learned tasks) available in CIL can be exploited to design a better and principled method for task-id prediction. We call the new method **TPL** (***T**ask-id **P**rediction based on **L**ikelihood Ratio*). TPL markedly outperforms strong CIL baselines and has **negligible catastrophic forgetting**.[1]

## 1 INTRODUCTION

Continual learning learns a sequence of tasks, $1, 2, \cdots, T$, incrementally (Ke & Liu, 2022; De Lange et al., 2021). Each task $t$ consists of a set of classes to be learned. This paper focuses on the challenging CL setting of *class-incremental learning* (CIL) (Rebuffi et al., 2017). The key challenge of CIL lies in the absence of task-identifier (task-id) in testing. There is another CL setting termed *task-incremental learning* (TIL), which learns a separate model or classifier for each task. In testing, the task-id is provided for each test sample so that it is classified by the task specific model.

A main assumption of continual learning is that once a task is learned, its training data is no longer accessible. This causes *catastrophic forgetting* (CF), which refers to performance degradation of previous tasks due to parameter updates in learning each new task (McCloskey & Cohen, 1989). An additional challenge specifically for CIL is *inter-task class separation* (ICS) (Kim et al., 2022b). That is, when learning a new task, it is hard to establish decision boundaries between the classes of the new task and the classes of the previous tasks without the training data of the previous tasks. Although in replay-based methods (Rebuffi et al., 2017; Kemker & Kanan, 2017; Lopez-Paz & Ranzato, 2017), a small number of training samples can be saved from each task (called the *replay data*) to help deal with CF and ICS to some extent by jointly training the new task data and the replay data from previous tasks, the effect on CF and ICS is limited as the number of replay samples is very small.

An emerging theoretically justified approach to solving CIL is to combine a TIL technique with an out-of-distribution (OOD) detection method, called the TIL+OOD approach (Kim et al., 2022b). The TIL method learns a model for each task in a shared network. The model for each task is not a

---

traditional classifier but an OOD detector. Note that almost all OOD detection methods can perform two tasks (1) *in-distribution* (IND) classification and (2) *out-of-distribution* (OOD) detection (Vaze et al., 2022). At test time, for each test sample, the system first computes a *task-id prediction* (TP) probability and a *within-task prediction* (WP) probability (Kim et al., 2022b) (same as IND classification) for each task. The two probabilities are then combined to make the final classification decision, which produces state-of-the-art results (Kim et al., 2022b; 2023). In this approach, WP is usually very accurate because it uses the task-specific model. **TP is the key challenge**.

There is a related existing approach that first predicts task-id and then predicts the class of the test sample using the task-specific model (Rajasegaran et al., 2020; Abati et al., 2020; Von Oswald et al., 2019). However, what is new is that Kim et al. (2022b) theoretically proved that TP is correlated with OOD detection of each task. Thus, the OOD detection capability of each task model can be used for task-id prediction of each test sample. The previous methods did not realize this and thus performed poorly (Kim et al., 2022b). In Kim et al. (2022b), the authors used the TIL method `HAT` (Serra et al., 2018) and OOD detection method `CSI` (Tack et al., 2020). `HAT` is a parameter isolation method for TIL, which learns a model for each task in a shared network and each task model is protected with learned masks to overcome CF. Each task model is an OOD detector based on `CSI`.[2]

Our paper argues that using traditional OOD detectors is not optimal for task-id prediction as they are not designed for CIL and thus do not exploit the information available in CIL for better task-id prediction. By leveraging the information in CIL, we can do much better. A new method for task-id prediction is proposed, which we call **TPL** (**T**ask-id **P**rediction based on **L**ikelihood Ratio). It consists of two parts: (1) a new method to train each task model and (2) a novel and principled method for task-id prediction, i.e., to estimate the probability of a test sample $x$ belonging to a task $t$, i.e., $\mathbf{P}(t|x)$). We formulate the estimation of $\mathbf{P}(t|x)$ as a binary selection problem between two events "$x$ belongs to $t$" and "$x$ belongs to $t^c$". $t^c$ is $t$'s complement with regard to the universal set $U_{CIL}$, which consists of all tasks that have been learned, i.e., $U_{CIL} = \{1, 2, \cdots, T\}$ and $t^c = U_{CIL} - \{t\}$.

The idea of TPL is analogous to using OOD detection for task-id prediction in the previous work. However, there is a **crucial difference**. In traditional OOD detection, given a set $U_{IND}$ of in-distribution classes, we want to estimate the probability that a test sample does not belong to any classes in $U_{IND}$. This means the universal set $U_{OOD}$ for OOD detection includes all possible classes in the world (except those in $U_{IND}$), which is at least very large if not infinite in size and we have no data from $U_{OOD}$. Then, there is no way we can estimate the distribution of $U_{OOD}$. However, we can estimate the distribution of $U_{CIL}$ based on the saved replay data[3] from each task in CIL. This allows us to use the *likelihood ratio* of $\mathcal{P}_t$ and $\mathcal{P}_{t^c}$ to provide a principled solution towards the binary selection problem and consequently to produce the task-id prediction probability $\mathbf{P}(t|x)$ as analyzed in Sec. 4.1, where $\mathcal{P}_t$ is the distribution of the data in task $t$ and $\mathcal{P}_{t^c}$ is the distribution of the data in $t^c$ (all other tasks than $t$), i.e., $t$'s complement ($t^c = U_{CIL} - \{t\}$).

The proposed system (also called TPL) uses the learned masks in the TIL method `HAT` for overcoming CF but the model for each task within `HAT` is not a traditional classifier but a model that facilitates task-id prediction (Sec. 3). At test time, given a test sample, the proposed likelihood ratio method is integrated with a logit-based score using an energy function to compute the task-id prediction probability and within-task prediction probability for the test sample to finally predict its class. Our experiments *with and without* using a pre-trained model show that TPL markedly outperforms strong baselines. With a pre-trained model, TPL has **almost no forgetting** or **performance deterioration**. We also found that the current formula for computing the ***forgetting rate*** is **not appropriate** for CIL.

## 2 RELATED WORK

**OOD Detection.** OOD detection has been studied extensively. Hendrycks & Gimpel (2016) use the maximum softmax probability (MSP) as the OOD score. Some researchers also exploit the logit space (Liang et al., 2017; Liu et al., 2020a; Sun et al., 2021), and the feature space to compute the distance from the test sample to the training data/IND distribution, e.g., Mahalanobis distance (Lee et al., 2018b) and KNN (Sun et al., 2022). Some use real/generated OOD data (Wang et al., 2022d; Liu et al., 2020a; Lee et al., 2018a). Our task-id prediction **does not use** any existing OOD method.

---

[2] In (Kim et al., 2023), it was also shown that based on this approach, CIL is learnable.

[3] In our case, the saved replay data are used to estimate the distribution of $U_{CIL}$ rather than to replay them in training a new task like replay-based methods. Also, our work is not about online continual learning.

**Continual Learning (CL).** Existing CL methods are of four main types. (1) *Regularization-based* methods address forgetting (CF) by using regularizers in the loss function (Kirkpatrick et al., 2017; Zhu et al., 2021) or orthogonal projection (Zeng et al., 2019) to preserve previous important parameters. The regularizers in DER (Yan et al., 2021) and BEEF (Wang et al., 2022a) are similar to OOD detection but they expand the network for each task and perform markedly poorer than our method. (2) *Replay-based* methods save a few samples from each task and replay them in training new tasks (Kemker & Kanan, 2017; Lopez-Paz & Ranzato, 2017; Li et al., 2022). However, replaying causes data imbalance (Guo et al., 2023; Xiang & Shlizerman, 2023; Ahn et al., 2021). (3) *Parameter isolation* methods train a sub-network for each task. HAT (Serra et al., 2018) and SupSup (Wortsman et al., 2020) are two representative methods. This approach is mainly used in task-incremental learning (TIL) and can eliminate CF. (4) *TIL+OOD* based methods have been discussed in Sec. 1.

Recently, using pre-trained models has become a standard practice for CL in both NLP (Ke et al., 2021a;b; 2023; Shao et al., 2023). and computer vision (CV) (Kim et al., 2022a; Wang et al., 2022e). See the surveys (Ke & Liu, 2022; Wang et al., 2023; De Lange et al., 2021; Hadsell et al., 2020).

Our work is closely related to CIL methods that employ a TIL technique and a task-id predictor. iTAML (Rajasegaran et al., 2020) assumes that each test batch is from a single task and uses the whole batch to detect the task-id. This assumption is unrealistic. CCG (Abati et al., 2020) uses a separate network to predict the task-id. Expert Gate (Aljundi et al., 2017) builds a distinct auto-encoder for each task. HyperNet (Von Oswald et al., 2019) and PR-Ent (Henning et al., 2021) use entropy to predict the task-id. However, these systems perform poorly as they did not realize that *OOD detection is the key to task-id prediction* (Kim et al., 2022b), which proposed the TIL+OOD approach. Kim et al. (2022b) gave two methods HAT+CSI and SupSup+CSI (Kim et al., 2022b). These two methods do not use a pre-trained model or replay data. The same approach was also taken in MORE (Kim et al., 2022a) and ROW (Kim et al., 2023) but they employ a pre-trained model and replay data in CIL. These methods have established a state-of-the-art performance. We have discussed how our proposed method TPL is different from them in the introduction section.

## 3 OVERVIEW OF THE PROPOSED METHOD

**Preliminary**. *Class incremental learning* (CIL) learns a sequence of tasks $1, ..., T$. Each task $t$ has an input space $\mathcal{X}^{(t)}$, a label space $\mathcal{Y}^{(t)}$, and a training set $\mathcal{D}^{(t)} = \{(\boldsymbol{x}_j^{(t)}, y_j^{(t)})\}_{j=1}^{n^{(t)}}$ drawn *i.i.d.* from $\mathcal{P}_{\mathcal{X}^{(t)}\mathcal{Y}^{(t)}}$. The class labels of the tasks are disjoint, i.e., $\mathcal{Y}^{(i)} \cap \mathcal{Y}^{(k)} = \emptyset, \forall i \neq k$. The goal of CIL is to learn a function $f : \cup_{t=1}^T \mathcal{X}^{(t)} \to \cup_{t=1}^T \mathcal{Y}^{(t)}$ to predict the class label of each test sample $\boldsymbol{x}$.

Kim et al. (2022b) proposed a theory for solving CIL. It decomposes the CIL probability of a test sample $\boldsymbol{x}$ of the $j$-th class $y_j^{(t)}$ in task $t$ into two probabilities (as the classes in all tasks are disjoint),

$$\mathbf{P}(y_j^{(t)}|\boldsymbol{x}) = \mathbf{P}(y_j^{(t)}|\boldsymbol{x}, t)\mathbf{P}(t|\boldsymbol{x}). \tag{1}$$

The two probabilities on the right-hand-side (R.H.S) define the CIL probability on the left-hand-side (L.H.S). The first probability on the R.H.S. is the ***within-task prediction*** (**WP**) probability and the second probability on the R.H.S. is the ***task-id prediction*** (**TP**) probability. Existing TIL+OOD methods basically use a traditional OOD detection method to build each task model. The OOD detection model for each task is exploited for estimating both TP and WP probabilities (see Sec. 1).

**Overview of the Proposed TPL.** This paper focuses on proposing a novel method for estimating task-id prediction probability, i.e., the probability of a test sample $\boldsymbol{x}$ belonging to (or drawing from the distribution of) a task $t$, i.e., $\mathbf{P}(t|\boldsymbol{x})$ in Sec. 1. The WP probability $\mathbf{P}(y_j^{(t)}|\boldsymbol{x}, t)$ can be obtained directly from the model of each task.

The mask-based method in HAT is used by our method to prevent CF. Briefly, in learning each task, it learns a model for the task and also a set of masks for those important neurons to be used later to prevent the model from being updated by future tasks. In learning a new task, the masks of previous models stop the gradient flow to those masked neurons in back-propagation, which eliminates CF. In the forward pass, all the neurons can be used, so the network is shared by all tasks. We note that our method can also leverage some other TIL methods other than HAT to prevent CF (see Appendix G).

The proposed method TPL is illustrated in Figure 1. It has two techniques for accurate estimation of $\mathbf{P}(t|\boldsymbol{x})$, one in training and one in testing (inference).

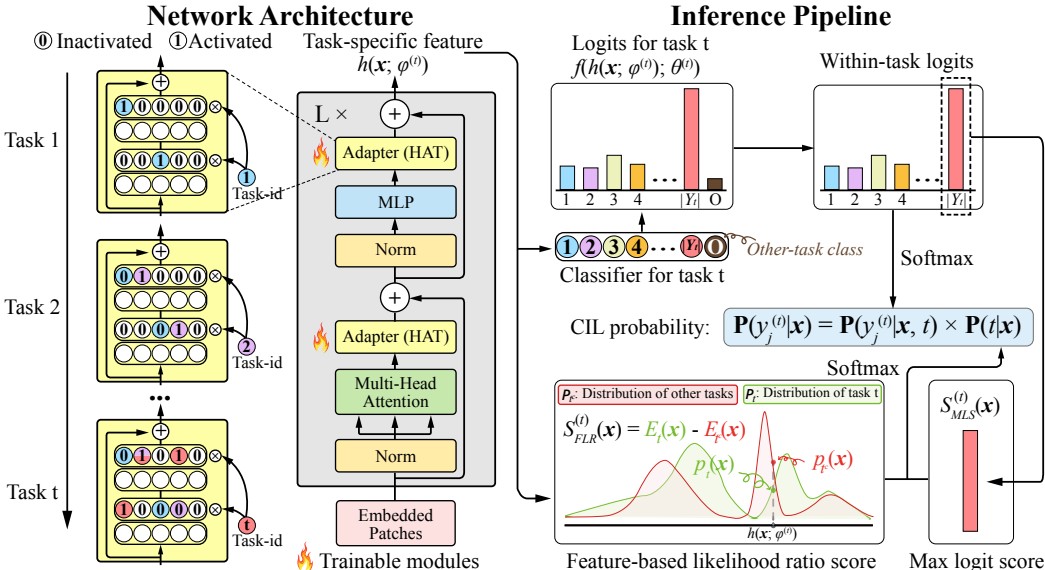

Figure 1: Illustration of the proposed TPL. We use a pre-trained transformer network (in the grey box) (see Sec. 5.1 for the case without using a pre-trained network). The pre-trained network is fixed and only the adapters (Houlsby et al., 2019) inserted into the transformer are trainable to adapt to specific tasks. **It is important to note that** the adapter (in yellow) used by HAT learns all tasks within the same adapter. The yellow boxes on the left show the progressive changes to the adapter as more tasks are learned.

(1) **Training**: In the original HAT, each model is a traditional supervised classifier trained with cross-entropy. However, for our purpose of predicting task-id, this is insufficient because it has no consideration of the other classes learned from other tasks. In TPL, each model for a task $t$ is trained using the classes $\mathcal{Y}^{(t)}$ of task $t$ and an extra class (called O, for others) representing the replay buffer data $Buf_{<t}$ of all the previous tasks. This enables each model to consider not only the new task data but also previous tasks' data, which facilitates more accurate computation of $\mathbf{P}(t|\boldsymbol{x})$.

For each task $t$, its model consists of a feature extractor $h(\boldsymbol{x}; \phi^{(t)})$ (partially shared with other tasks based on HAT), and a task-specific classifier $f(\boldsymbol{z}; \theta^{(t)})$. When learning task $t$, the model receives the training data $\mathcal{D}^{(t)}$ and the replay data $Buf_{<t}$ (stored in a memory buffer). Then we minimize the loss:

$$\mathcal{L}(\theta^{(t)}, \phi^{(t)}) = \mathbb{E}_{(\boldsymbol{x},y)\sim\mathcal{D}^{(t)}\cup Buf_{<t}} \left[ \mathcal{L}_{CE}(f(h(\boldsymbol{x}; \phi^{(t)}); \theta^{(t)}), y) \right] + \mathcal{L}_{HAT}, \quad (2)$$

where $\mathcal{L}_{CE}$ is the cross-entropy loss, $\mathcal{L}_{HAT}$ is the regularization loss used in HAT (see Appendix G).

(2) **Testing (or inference)**: We follow eq. (1) to compute the CIL probability. The WP probability $(\mathbf{P}(y_j^{(t)}|\boldsymbol{x}, t))$ for each test sample is computed through softmax on only the original classes $\mathcal{Y}^{(t)}$ of task $t$, the first term on the right of eq. (3) (also see the top right part in Figure 1). The O class is not used in inference. Note that the probabilities for different tasks can be computed in parallel.

$$\mathbf{P}(y_j^{(t)}|\boldsymbol{x}) = \left[ softmax\left( f(h(\boldsymbol{x}; \phi^{(t)}); \theta^{(t)}) \right) \right]_j \cdot \mathbf{P}(t|\boldsymbol{x}) \quad (3)$$

The class $y_j^{(t)}$ with the highest probability will be predicted as the class for test sample $\boldsymbol{x}$.[4] We discuss the proposed method for computing task-id prediction probability $\mathbf{P}(t|\boldsymbol{x})$ (see the bottom right part in Figure 1) in the next section. Training will not be discussed any further.

## 4 ESTIMATING TASK-ID PREDICTION PROBABILITY

### 4.1 THEORETICAL ANALYSIS

As noted in Sec. 1, we estimate the TP probability $\mathbf{P}(t|\boldsymbol{x})$ by predicting whether a sample $\boldsymbol{x}$ is drawn from the distribution $\mathcal{P}_t$ of task $t$ or drawn from the distribution of $t$'s complement $t^c$,

---

[4]We also calibrate the probabilities from different task models, but it has little effect (see Appendix B).

i.e., $\mathcal{P}_{t^c}$. We denote the universal set $U_{CIL}$ of all tasks (or task-ids) that have been learned, i.e., $U_{CIL} = \{1, 2, \cdots, T\}$ and $t^c = U_{CIL} - \{t\}$. From a frequentist perspective, our objective can be formulated as a binary hypothesis test:

$$\mathcal{H}_0 : \boldsymbol{x} \sim \mathcal{P}_t \quad v.s. \quad \mathcal{H}_1 : \boldsymbol{x} \sim \mathcal{P}_{t^c}, \tag{4}$$

Using the Neyman-Pearson lemma (Neyman & Pearson, 1933), we can derive a theorem that demonstrates the principled role of likelihood ratio in this task (the proofs are given in Appendix E):

**Theorem 4.1** *A test with rejection region $\mathcal{R}$ defined as follows is a unique **uniformly most powerful (UMP) test** for the hypothesis test problem defined in eq. (4):*

$$\mathcal{R} := \{\boldsymbol{x} : p_t(\boldsymbol{x})/p_{t^c}(\boldsymbol{x}) < \lambda_0\}.$$

*where $\lambda_0$ is a threshold that can be chosen to obtain a specified significance level.*

**Theorem 4.2** *The UMP test for hypothesis test defined in eq. (4) maximizes the Area Under the Curve (AUC) of binary classification between $\mathcal{P}_t$ and $\mathcal{P}_{t^c}$.*

Theorems 4.1 and 4.2 highlight the importance of detecting samples that do not belong to task $t$ based on low $t$ density $p_t(\boldsymbol{x})$ and high $t^c$ density $p_{t^c}(\boldsymbol{x})$.

Note that in traditional OOD detection, the system has no access to the true OOD distribution $\mathcal{P}_{t^c}$ but only $\mathcal{P}_t$ (IND distribution). Some existing methods resort to a proxy distribution $\mathcal{P}_{t^c}^{proxy}$, such as a uniform distribution (Nalisnick et al., 2018) or an auxiliary data distribution (Lin & Gu, 2023) as the universal set $U$ is the set of all classes in the world and the universal set of all OOD classes for task $t$ denoted by $U_{OOD}^{(t)}$ is very large if not infinite. This approach can lead to potential risks. For instance, consider a scenario where $\mathcal{P}_{t^c} = \mathcal{N}(0, 0.01)$ and $\mathcal{P}_t = \mathcal{N}(0, 1)$. It is apparent that $p_t(0) > p_t(1)$, but 0 is more likely to belongs to $\mathcal{P}_{t^c}$ than 1 as $0.1 = p_t(0)/p_{t^c}(0) < p_t(1)/p_{t^c}(1) = 0.1 \cdot e^{49.5}$. We further show the failure cases in real CIL scenarios in Appendix H.

**Good News for CIL.** In CIL, the IND distribution $\mathcal{P}_t$ for task $t$ can be interpreted as the marginal distribution $\mathcal{P}_{\mathcal{X}^{(t)}}$, while $\mathcal{P}_{t^c}$ corresponds to a mixture distribution $\mathcal{P}_{\mathcal{X}^{(t^c)}}$ comprising the individual marginal distributions $\{\mathcal{P}_{\mathcal{X}^{(t^*)}}\}_{t^* \neq t}$ (which can be estimated based on the saved replay data), each of which is assigned the equal mixture weight. Consequently, we have the knowledge of $\mathcal{P}_{t^c}$ in CIL, thereby offering an opportunity to estimate $\mathcal{P}_{t^c}$ to be used to compute task-id prediction $\mathbf{P}(t|\boldsymbol{x})$ more accurately. This leads to our design of `TPL` in the following subsections.

## 4.2 COMPUTING TASK-ID PREDICTION PROBABILITY

We now present the proposed method for computing the task-id prediction probability $\mathbf{P}(t|\boldsymbol{x})$, which has three parts: (1) estimating both $\mathcal{P}_t$ and $\mathcal{P}_{t^c}$ (as analyzed in Sec. 4.1) and computing the likelihood ratio, (2) integrating the likelihood ratio based score with a logit-based score for further improvement, and (3) applying a softmax function on the scores for all tasks to obtain the task-id prediction probability for each task. The three parts correspond to the bottom right part of Figure 1.

### 4.2.1 ESTIMATING $\mathcal{P}_t$ AND $\mathcal{P}_{t^c}$ AND COMPUTING LIKELIHOOD RATIO

Guided by Theorem 4.1, we design a task-id prediction score based on the likelihood ratio $p_t(\boldsymbol{x})/p_{t^c}(\boldsymbol{x})$. However, due to the challenges in directly estimating the data distribution within the high-dimensional raw image space, we instead consider estimation in the low-dimensional *feature space*. Interestingly, many distance-based OOD detection scores can function as density estimators that estimate the IND density $p(\boldsymbol{x})$ in the *feature space* (see Appendix E.4 for justifications). For instance, `MD` (*Mahalanobis Distance*) (Lee et al., 2018b) estimates distributions using Gaussian mixture models, while `KNN` (Sun et al., 2022) uses non-parametric estimation. Our method `TPL` also uses the two scores to estimate distributions (i.e., $\mathcal{P}_t$ and $\mathcal{P}_{t^c}$ in our case).

To connect the normalized probability density with unnormalized task-id prediction scores, we leverage energy-based models (EBMs) to parameterize $\mathcal{P}_t$ and $\mathcal{P}_{t^c}$. Given a test sample $\boldsymbol{x}$, it has density $p_t(\boldsymbol{x}) = \exp\{E_t(\boldsymbol{x})\}/Z_1$ in $\mathcal{P}_t$, and density $p_{t^c}(\boldsymbol{x}) = \exp\{E_{t^c}(\boldsymbol{x})\}/Z_2$ in $\mathcal{P}_{t^c}$, where $Z_1, Z_2$ are normalization constants that ensure the integral of densities $p_t(\boldsymbol{x})$ and $p_{t^c}(\boldsymbol{x})$ equal 1,

and $E_t(\cdot)$, $E_{t^c}(\cdot)$ are called *energy functions*.[5] Consequently, we can design a feature-based task-id prediction score using the **L**ikelihood **R**atio (LR), which is also shown at the bottom right of Figure 1:

$$S_{LR}^{(t)}(\boldsymbol{x}) = \log(p_t(\boldsymbol{x})/p_{t^c}(\boldsymbol{x})) = E_t(\boldsymbol{x}) - E_{t^c}(\boldsymbol{x}) + \log(Z_2/Z_1). \tag{5}$$

Since $\log(Z_2/Z_1)$ is a constant, it can be omitted in the task-id prediction score definition:

$$S_{LR}^{(t)}(\boldsymbol{x}) := E_t(\boldsymbol{x}) - E_{t^c}(\boldsymbol{x}), \tag{6}$$

Since the energy functions $E_t(\cdot)$ and $E_{t^c}(\cdot)$ need not to be normalized, we estimate them with the above scores. We next discuss how to choose specific $E_t(\cdot)$ and $E_{t^c}(\cdot)$ for eq. (6).

For in-task energy $E_t(\boldsymbol{x})$ of a task, we simply adopt an OOD detection score $S_{MD}(\boldsymbol{x})$, which is the OOD score for MD and is defined as the inverse of the minimum Mahalanobis distance of feature $h(\boldsymbol{x}; \phi^{(t)})$ to all class centroids. The details of how $S_{MD}^{(t)}(\boldsymbol{x})$ is computed are given in Appendix F.1.

For out-of-task energy $E_{t^c}(\boldsymbol{x})$ of a task, we use **replay data** from other tasks for estimation. Let $Buf_{t^c}$ be the set of buffer/replay data excluding the data of classes in task $t$. We set $E_{t^c}(\boldsymbol{x}) = -d_{KNN}(\boldsymbol{x}, Buf_{t^c})$, where $d_{KNN}(\boldsymbol{x}, Buf_{t^c})$ is the $k$-nearest distances of the feature $h(\boldsymbol{x}; \phi)^{(t)}$ to the set of features of the replay $Buf_{t^c}$ data. If $d_{KNN}(\boldsymbol{x}, Buf_{t^c})$ is small, it means the distance between $\boldsymbol{x}$ and replay $Buf_{t^c}$ data is small in the feature space. The vanilla KNN score is $S_{KNN}^{(t)}(\boldsymbol{x}) = -d_{KNN}(\boldsymbol{x}, \mathcal{D}^{(t)})$, which was originally designed to estimate $p_t(\boldsymbol{x})$ using the training set $\mathcal{D}^{(t)}$. Here we adopt it to estimate $p_{t^c}(\boldsymbol{x})$ using the replay data ($Buf_{t^c}$). Finally, we obtain,

$$S_{LR}^{(t)}(\boldsymbol{x}) := \underbrace{\alpha \cdot S_{MD}^{(t)}(\boldsymbol{x})}_{E_t(\boldsymbol{x})} + \underbrace{d_{KNN}(\boldsymbol{x}, Buf_{t^c})}_{-E_{t^c}(\boldsymbol{x})}, \tag{7}$$

where $\alpha$ is a hyper-parameter to make the two scores comparable. This is a principled task-id prediction score as justified in Sec. 4.1.

**Remarks.** We can also use some other feature-based estimation methods instead of MD and KNN in $S_{LR}(\boldsymbol{x})$. The reason why we choose MD to estimate $\mathcal{P}_t$ is that it does not require the task data at test time (but KNN does), and we choose KNN to estimate $\mathcal{P}_{t^c}$ because the non-parametric estimator KNN is high performing (Yang et al., 2022) and we use only the saved replay data for this. We will conduct an ablation study using different estimation methods for both $\mathcal{P}_t$ and $\mathcal{P}_{t^c}$ in Sec. 5.3.

### 4.2.2 COMBINING WITH A LOGIT-BASED SCORE

To further improve the task-id prediction score, we combine the feature-based $S_{LR}$ score with a logit-based score, which has been shown quite effective in OOD detection (Wang et al., 2022c).

We again develop an energy-based model (EBM) framework for the combination that offers a principled approach to composing different task-id prediction scores. Specifically, to combine the proposed feature-based $S_{LR}^{(t)}(\cdot)$ score with a logit-based score (an energy function) $S_{logit}^{(t)}(\cdot)$, we can make the composition as:

$$E_{composition}(\boldsymbol{x}) = \log(\exp\{\alpha_1 \cdot S_{logit}^{(t)}(\boldsymbol{x})\} + \exp\{\alpha_2 \cdot S_{LR}^{(t)}(\boldsymbol{x})\}), \tag{8}$$

where $\alpha_1$ and $\alpha_2$ are scaling terms to make different scores comparable. As noted in (Du et al., 2020), the composition emulates an OR gate for energy functions.

To choose a logit-based method for $S_{logit}^{(t)}(\cdot)$ in eq. (8), we opt for the simple yet effective method MLS score $S_{MLS}^{(t)}(\boldsymbol{x})$, which is defined as the *maximum logit* of $\boldsymbol{x}$ (also shown on the right of Figure 1).

Our final score $S_{TPL}^{(t)}(\boldsymbol{x})$, which integrates feature-based $S_{LR}^{(t)}(\cdot)$ and the logit-based $S_{MLS}^{(t)}(\cdot)$ scores, uses the composition in Eq. 8:

$$S_{TPL}^{(t)}(\boldsymbol{x}) = \log\left(\exp\{\beta_1 \cdot S_{MLS}^{(t)}(\boldsymbol{x})\} + \exp\{\beta_2 \cdot S_{MD}^{(t)}(\boldsymbol{x}) + d_{KNN}(\boldsymbol{x}, Buf_{t_c})\}\right), \tag{9}$$

---

[5] In EBMs, the density $p(x)$ is typically defined as $\exp\{-E(x)\}/Z$. Since our task-id prediction score is defined to measure the likelihood that the test sample belongs to a task, the energy function here is defined as positively related to the probability density.

where $\beta_1$ and $\beta_2$ are scaling terms, which are given by merging $\alpha$ in eq. (7) and $\alpha_1$, $\alpha_2$ in eq. (8). Since the scale of $d_{KNN}(\cdot)$ is near to 1, we simply choose $\beta_1$ and $\beta_2$ to be the inverse of empirical means of $S_{MLS}^{(t)}(\boldsymbol{x})$ and $S_{MD}^{(t)}(\boldsymbol{x})$ estimated by the training data $\mathcal{D}^{(t)}$ to make different scores comparable:

$$\frac{1}{\beta_1} = \frac{1}{|\mathcal{D}^{(t)}|} \sum_{\boldsymbol{x} \in \mathcal{D}^{(t)}} S_{MLS}^{(t)}(\boldsymbol{x}), \quad \frac{1}{\beta_2} = \frac{1}{|\mathcal{D}^{(t)}|} \sum_{\boldsymbol{x} \in \mathcal{D}^{(t)}} S_{MD}^{(t)}(\boldsymbol{x}) \tag{10}$$

**Remarks**. We exploit **EBMs**, which are known for their *flexibility* but suffering from *intractability*. However, we exploit EBMs' *flexibility* to derive principled task-id prediction score following Theorem 4.1 and eq. (8), while keeping the *tractability* via approximation using OOD scores (MD, KNN, MLS) in practice. This makes our proposed TPL maintain both theoretical and empirical soundness.

### 4.3 CONVERTING TASK-ID PREDICTION SCORES TO PROBABILITIES

Although theoretically principled as shown in Sec. 4.1, our final task-id prediction score is still an unnormalized energy function. We convert the task-id prediction scores for all tasks (i.e., $\{S_{TPL}^{(t)}(\boldsymbol{x})\}_{t=1}^T$) to normalized probabilities via softmax:

$$\mathbf{P}(t|\boldsymbol{x}) = softmax\left(\left[S_{TPL}^{(1)}(\boldsymbol{x}), S_{TPL}^{(2)}(\boldsymbol{x}), \cdots, S_{TPL}^{(T)}(\boldsymbol{x})\right] / \gamma\right)_t, \tag{11}$$

where $\gamma$ is a temperature parameter. To encourage confident task-id prediction, we set a low temperature $\gamma = 0.05$ to produce a low entropy task-id preidction distribution for all our experiments.

## 5 EXPERIMENTS

### 5.1 EXPERIMENTAL SETUP

**CIL Baselines.** We use **17 *baselines***, including **11 *replay methods***: iCaRL (Rebuffi et al., 2017), A-GEM (Chaudhry et al., 2018), EEIL (Castro et al., 2018), GD (Lee et al., 2019), DER++ (Buzzega et al., 2020), HAL (Chaudhry et al., 2021), DER (Yan et al., 2021), FOSTER (Wang et al., 2022b), AFC (Kang et al., 2022), BEEF (Wang et al., 2022a), MORE (Kim et al., 2022a), ROW (Kim et al., 2023), and **6 *non-replay methods***: HAT (Serra et al., 2018), ADAM (Zhou et al., 2023), OWM (Zeng et al., 2019), PASS (Zhu et al., 2021), SLDA (Hayes & Kanan, 2020), and L2P (Wang et al., 2022e).[6] We follow (Kim et al., 2022b) to adapt HAT (which is a TIL method) for CIL and call it $\text{HAT}_{CIL}$. **Implementation details, network size and running time** are given in Appendix I.1.

**Datasets.** To form a sequence of tasks in CIL experiments, we follow the common CIL setting. We split CIFAR-10 into 5 tasks (2 classes per task) (**C10-5T**). For CIFAR-100, we conduct two experiments: 10 tasks (10 classes per task) (**C100-10T**) and 20 tasks (5 classes per task) (**C100-20T**). For TinyImageNet, we split 200 classes into 5 tasks (40 classes per task) (**T-5T**) and 10 tasks (20 classes per task) (**T-10T**). We set the replay buffer size for CIFAR-10 as 200 samples, and CIFAR-100 and TinyImageNet as 2000 samples following Kim et al. (2023). Following the random class order protocol in Rebuffi et al. (2017), we randomly generate five different class orders for each experiment and report the averaged metrics over the 5 random orders. For a fair comparison, the class orderings are kept the same for all systems. Results on a larger dataset are given in Appendix D.1.

**Backbone Architectures.** We conducted two sets of experiments, one **using a pre-trained model** and one **without using a pre-trained model**. Here we focus on using a pre-trained model as that is getting more popular. Following the TIL+OOD works (Kim et al., 2022a; 2023), TPL uses the same DeiT-S/16 model (Touvron et al., 2021) pre-trained using 611 classes of ImageNet after removing 389 classes that are similar or identical to the classes of the experiment data CIFAR and TinyImageNet to prevent information leak (Kim et al., 2022a; 2023). To leverage the pre-trained model while adapting to new knowledge, we insert an adapter module (Houlsby et al., 2019) at each transformer layer except SLDA and L2P.[7] The adapter modules, classifiers, and layer norms are trained using HAT while the transformer parameters are fixed to prevent CF. The hidden dimension of adapters is 64

---

[6] The systems HAT+CSI and Sup+CSI in (Kim et al., 2022b) (which are based on the TIL+OOD paradigm but do not use a pre-trained model) are not included as they are much weaker because their contrastive learning and data augmentations do not work well with a pre-trained model.

[7] SLDA fine-tunes only the classifier with a fixed feature extractor and L2P trains learnable prompts.

Table 1: CIL ACC (%). "-XT": X number of tasks. The best result in each column is highlighted in bold. The baselines are divided into two groups via the dashed line. The first group contains non-replay methods, and the second group contains replay-based methods. **Non-CL** (non-continual learning) denotes pooling all tasks together to learn all classes as one task, which gives the performance **upper bound** for CIL. **AIA** is the *average incremental* ACC (%). **Last** is the ACC after learning the final task. See **forgetting rate results** in Appendix C.2. The pink rows also show the results of Non-CL$_{PFI}$ and TPL$_{PFI}$, which use DeiT Pre-trained with **Full I**mageNet.

| | C10-5T | | C100-10T | | C100-20T | | T-5T | | T-10T | | Average | |
| | Last | AIA | Last | AIA | Last | AIA | Last | AIA | Last | AIA | Last | AIA |
|---|---|---|---|---|---|---|---|---|---|---|---|---|
| **Non-CL** | $95.79^{\pm0.15}$ | $97.01^{\pm0.14}$ | $82.76^{\pm0.22}$ | $87.20^{\pm0.29}$ | $82.76^{\pm0.22}$ | $87.53^{\pm0.31}$ | $72.52^{\pm0.41}$ | $77.03^{\pm0.47}$ | $72.52^{\pm0.41}$ | $77.03^{\pm0.41}$ | 81.27 | 85.16 |
| OWM | $41.69^{\pm6.34}$ | $56.00^{\pm3.46}$ | $21.39^{\pm3.18}$ | $40.10^{\pm1.86}$ | $16.98^{\pm4.44}$ | $32.58^{\pm1.58}$ | $24.55^{\pm2.48}$ | $45.18^{\pm0.33}$ | $17.52^{\pm3.45}$ | $35.75^{\pm2.21}$ | 24.43 | 41.92 |
| ADAM | $83.92^{\pm0.51}$ | $90.33^{\pm0.42}$ | $61.21^{\pm0.36}$ | $72.55^{\pm0.41}$ | $58.99^{\pm0.61}$ | $70.89^{\pm0.51}$ | $50.11^{\pm0.46}$ | $61.85^{\pm0.51}$ | $49.68^{\pm0.40}$ | $61.44^{\pm0.44}$ | 60.78 | 71.41 |
| PASS | $86.21^{\pm1.10}$ | $89.03^{\pm7.13}$ | $68.90^{\pm0.94}$ | $77.01^{\pm2.44}$ | $66.77^{\pm1.18}$ | $76.42^{\pm1.23}$ | $61.03^{\pm0.38}$ | $67.12^{\pm6.26}$ | $58.34^{\pm0.42}$ | $67.33^{\pm3.63}$ | 68.25 | 75.38 |
| HAT$_{CIL}$ | $82.40^{\pm0.12}$ | $91.06^{\pm0.36}$ | $62.91^{\pm0.24}$ | $73.99^{\pm0.86}$ | $59.54^{\pm0.41}$ | $69.12^{\pm1.06}$ | $59.22^{\pm0.10}$ | $69.38^{\pm1.14}$ | $54.03^{\pm0.21}$ | $65.63^{\pm1.64}$ | 63.62 | 73.84 |
| SLDA | $88.64^{\pm0.05}$ | $93.54^{\pm0.66}$ | $67.82^{\pm0.05}$ | $77.72^{\pm0.58}$ | $67.80^{\pm0.05}$ | $78.51^{\pm0.58}$ | $57.93^{\pm0.05}$ | $66.03^{\pm1.35}$ | $57.93^{\pm0.06}$ | $67.39^{\pm1.81}$ | 68.02 | 76.64 |
| L2P | $73.59^{\pm4.15}$ | $84.60^{\pm2.28}$ | $61.72^{\pm0.81}$ | $72.88^{\pm1.18}$ | $53.84^{\pm1.59}$ | $66.52^{\pm1.61}$ | $59.12^{\pm0.96}$ | $67.81^{\pm1.25}$ | $54.09^{\pm1.14}$ | $64.59^{\pm1.59}$ | 60.47 | 71.28 |
| iCaRL | $87.55^{\pm0.99}$ | $89.74^{\pm6.63}$ | $68.90^{\pm0.47}$ | $76.50^{\pm3.56}$ | $69.15^{\pm0.99}$ | $77.06^{\pm2.36}$ | $53.13^{\pm1.04}$ | $61.36^{\pm6.21}$ | $51.88^{\pm2.36}$ | $63.56^{\pm3.08}$ | 66.12 | 73.64 |
| A-GEM | $56.33^{\pm7.77}$ | $68.19^{\pm3.24}$ | $25.21^{\pm4.00}$ | $43.83^{\pm0.69}$ | $21.99^{\pm4.01}$ | $35.97^{\pm1.15}$ | $30.53^{\pm3.99}$ | $49.26^{\pm0.64}$ | $21.90^{\pm5.52}$ | $39.58^{\pm3.32}$ | 31.19 | 47.37 |
| EEIL | $82.34^{\pm3.13}$ | $90.50^{\pm0.72}$ | $68.08^{\pm0.51}$ | $81.10^{\pm0.37}$ | $63.79^{\pm0.66}$ | $79.54^{\pm0.69}$ | $53.34^{\pm0.54}$ | $66.63^{\pm0.40}$ | $50.38^{\pm0.97}$ | $66.54^{\pm0.61}$ | 63.59 | 76.86 |
| GD | $89.16^{\pm0.53}$ | $94.22^{\pm0.75}$ | $64.36^{\pm0.57}$ | $80.51^{\pm0.57}$ | $60.10^{\pm0.74}$ | $78.43^{\pm0.76}$ | $53.01^{\pm0.97}$ | $67.51^{\pm0.38}$ | $42.48^{\pm2.53}$ | $63.91^{\pm0.40}$ | 61.82 | 76.92 |
| DER++ | $84.63^{\pm2.91}$ | $89.01^{\pm6.29}$ | $69.73^{\pm0.99}$ | $80.64^{\pm2.74}$ | $70.03^{\pm1.46}$ | $81.72^{\pm1.76}$ | $55.84^{\pm2.21}$ | $66.55^{\pm3.73}$ | $54.20^{\pm3.28}$ | $67.14^{\pm1.40}$ | 66.89 | 77.01 |
| HAL | $84.38^{\pm2.70}$ | $87.00^{\pm7.27}$ | $67.17^{\pm1.50}$ | $77.42^{\pm2.73}$ | $67.37^{\pm1.45}$ | $77.85^{\pm1.71}$ | $52.80^{\pm2.37}$ | $65.31^{\pm3.68}$ | $55.25^{\pm3.60}$ | $64.48^{\pm1.45}$ | 65.39 | 74.41 |
| DER | $86.79^{\pm1.20}$ | $92.83^{\pm1.10}$ | $73.30^{\pm0.58}$ | $82.89^{\pm0.45}$ | $72.00^{\pm0.57}$ | $82.79^{\pm0.76}$ | $59.57^{\pm0.89}$ | $70.32^{\pm0.57}$ | $57.18^{\pm1.40}$ | $70.21^{\pm0.86}$ | 69.77 | 79.81 |
| FOSTER | $86.09^{\pm0.38}$ | $91.54^{\pm0.65}$ | $71.69^{\pm0.24}$ | $81.16^{\pm0.39}$ | $72.91^{\pm0.45}$ | $83.02^{\pm0.86}$ | $54.44^{\pm0.28}$ | $69.95^{\pm0.28}$ | $55.70^{\pm0.40}$ | $70.00^{\pm0.26}$ | 68.17 | 79.13 |
| BEEF | $87.10^{\pm1.38}$ | $93.10^{\pm1.21}$ | $72.09^{\pm0.33}$ | $81.91^{\pm0.58}$ | $71.88^{\pm0.54}$ | $81.45^{\pm0.74}$ | $61.41^{\pm0.83}$ | $71.21^{\pm0.57}$ | $58.16^{\pm0.60}$ | $71.16^{\pm0.82}$ | 70.13 | 79.77 |
| MORE | $89.16^{\pm0.96}$ | $94.23^{\pm0.82}$ | $70.23^{\pm2.27}$ | $81.24^{\pm1.24}$ | $70.53^{\pm1.09}$ | $81.59^{\pm0.98}$ | $64.97^{\pm1.28}$ | $74.03^{\pm1.61}$ | $63.06^{\pm1.26}$ | $72.74^{\pm1.04}$ | 71.59 | 80.77 |
| ROW | $90.97^{\pm0.19}$ | $94.45^{\pm0.21}$ | $74.72^{\pm0.48}$ | $82.87^{\pm0.41}$ | $74.60^{\pm0.12}$ | $83.12^{\pm0.23}$ | $65.11^{\pm1.97}$ | $74.16^{\pm1.34}$ | $63.21^{\pm2.53}$ | $72.91^{\pm2.12}$ | 73.72 | 81.50 |
| **TPL (ours)** | $92.33^{\pm0.32}$ | $95.11^{\pm0.44}$ | $76.53^{\pm0.27}$ | $84.10^{\pm0.34}$ | $76.34^{\pm0.38}$ | $84.46^{\pm0.28}$ | $68.64^{\pm0.44}$ | $76.77^{\pm0.23}$ | $67.20^{\pm0.51}$ | $75.72^{\pm0.37}$ | 76.21 | 83.23 |
| Non-CL$_{PFI}$ | $96.90^{\pm0.07}$ | $97.96^{\pm0.05}$ | $83.61^{\pm0.33}$ | $89.72^{\pm0.10}$ | $83.61^{\pm0.33}$ | $88.89^{\pm0.06}$ | $85.55^{\pm0.07}$ | $88.26^{\pm0.08}$ | $85.71^{\pm0.14}$ | $88.66^{\pm0.01}$ | 87.08 | 90.70 |
| TPL$_{PFI}$ | $94.86^{\pm0.02}$ | $96.89^{\pm0.02}$ | $82.43^{\pm0.12}$ | $88.28^{\pm0.17}$ | $80.86^{\pm0.07}$ | $87.32^{\pm0.07}$ | $84.06^{\pm0.11}$ | $87.19^{\pm0.11}$ | $83.87^{\pm0.07}$ | $87.40^{\pm0.16}$ | 85.22 | 89.42 |

for CIFAR-10, and 128 for CIFAR-100 and TinyImageNet. For completeness, we also report the results of TPL using DeiT-S/16 **P**re-trained with the **Full I**mageNet (called **TPL$_{PFI}$**) in the pink rows of Table 1. The results without using a pre-trained model are given in Appendix D.2.

**Evaluation Metrics.** We use threepopular metrics: (1) *accuracy after learning the final task* (**Last** in Table 1), (2) *average incremental accuracy* (**AIA** in Table 1), and (3) *forgetting rate* (see Table 6 in Appendix C.2, where we also discuss why the **current forgetting rate formula** is **not appropriate** for CIL, but only for TIL. The definitions of all these metrics are given in Appendix C.

## 5.2 RESULTS AND COMPARISONS

Table 1 shows the CIL accuracy (ACC) results. The last two columns give the row averages. Our `TPL` performs the best in both *average incremental ACC* (AIA) and ACC *after the last task* (Last). Based on AIA, TPL's forgetting (CF) is almost negligible. When the full ImageNet data is used in pre-training (pink rows), TPL$_{PFI}$ has **almost no forgetting** in both AIA and Last ACC.

**Comparison with CIL baselines with pre-training.** The best-performing replay-based baseline is `ROW`, which also follows the TIL+OOD paradigm (Kim et al., 2022b). Since its OOD score is inferior to our principled $S_{LR}(x)$, ROW is greatly outperformed by TPL. The ACC gap between our `TPL` and the best exemplar-free method `PASS` is even greater, 68.25% (PASS) vs. 76.21% (TPL) in Last ACC. `TPL` also markedly outperforms the strong *network expansion* methods `DER`, `FOSTER`, and `BEEF`.

**Without pre-training.** The accuracy results after learning the final task without pre-training are given in Table 8 of Appendix D.2. We provide a summary in Table 2 here. As `L2P`, `SLDA`, and `ADAM` are designed specifically for pre-trained backbones, they cannot be adapted to the non-pre-training setting and thus are excluded here. Similar to the observation in Table 1, our `TPL` achieves the overall best results (with ACC of 57.5%), while `DER` ranks the second (54.2%).

Table 2: CIL ACC (%) after learning the final task without pre-training (average over the five datasets used in Table 1). The detailed results are shown in Table 8 of Appendix D.2.

| OWM | PASS | EEIL | GD | HAL | A-GEM | HAT | iCaRL |
|---|---|---|---|---|---|---|---|
| 24.7 | 30.6 | 46.3 | 47.1 | 46.4 | 45.8 | 39.5 | 45.6 |

| DER++ | DER | FOSTER | BEEF | MORE | ROW | **TPL (ours)** | |
|---|---|---|---|---|---|---|---|
| 46.5 | 54.2 | 52.2 | 53.4 | 51.2 | 53.1 | **57.5** | |

## 5.3 ABLATION STUDY

**Performance gain**. Figure 2(a) shows the performance gain achieved by adding each proposed technique. Starting from vanilla HAT$_{CIL}$ with an average Last ACC of 63.41% over all datasets, the proposed likelihood ratio LR score (HAT+LR) boosts the average Last ACC to 71.25%. Utilizing the OOD detection method MLS (HAT+MLS) only improves the ACC to 68.69%. The final composition of LR and MLS boosted the performance to 76.21%.

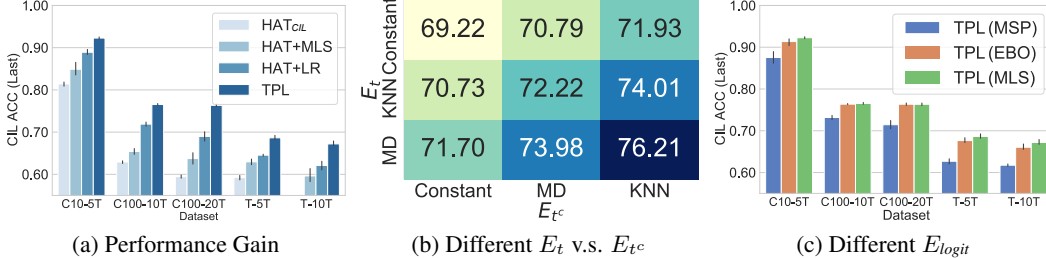

Figure 2: Ablation Studies. Fig (a) illustrates the achieved ACC gain for each of the designed techniques on the five datasets; Fig (b) displays the average ACC results obtained from different choices of $E_t$ and $E_{t^c}$ for eq. (7); Fig (c) showcases the results for various selections of $E_{logit}$ for TPL in eq. (9).

**Different $E_t$ v.s. $E_{t^c}$.** Recall that the key insight behind the LR score lies in the estimation of likelihood ratio. Figure 2(b) presents the average Last ACC results across 5 datasets, employing various approaches to estimate $\mathcal{P}_t$ and $\mathcal{P}_{t^c}$. In this context, the term *Constant* refers to the use of a uniform distribution as the distribution of $\mathcal{P}_{t^c}$, where the energy function is a constant mapping. Our TPL approach is equivalent to employing ($E_t = $ MD, $E_{t^c} = $ KNN). The results reveal the following: (1) The incorporation of the $\mathcal{P}_{t^c}$ distribution estimation is beneficial compared to assuming a uniform distribution. (2) As $\mathcal{P}_{t^c}$ can only be estimated using the replay data, the high-performing KNN method outperforms MD. However, since MD can estimate $\mathcal{P}_t$ without task $t$'s training data during the test phase, it proves to be more effective than KNN when serving as $E_t$.

**Different logit-based scores.** Although $S_{MLS}(\boldsymbol{x})$ is used as the logit-based score in Section 4.2.2, alternative logit-based scores can also be considered. In this study, we conduct experiments using 3 popular logit-based scores MSP (Hendrycks & Gimpel, 2016), EBO (Liu et al., 2020a), and MLS (their definitions are given in Appendix F.2). The results presented in Figure 2(c) indicate that EBO and MLS yield comparable results, with average Last ACC of 75.76%, and 76.21% respectively, while MSP has inferior performance with average Last ACC of 71.32%.

**Smaller replay buffer sizes.** The accuracy after learning the final task with smaller replay buffer sizes are given in Table 9 of Appendix D.3. We provide a summary as Table 3, which shows that when using a smaller replay buffer, the performance drop of TPL is small. The goal of using the replay data in TPL is to compute the likelihood ratio (LR) score, while traditional replay methods focus on preventing forgetting (CF). Note that CF is already addressed by the TIL method HAT in our case. Thus our method TPL is robust with fewer replay samples.

Table 3: ACC (%) after learning the final task (Last) with smaller replay buffer sizes (average over the five datasets in Table 1). The detailed results are shown in Table 9 of Appendix D.3. The replay buffer size is set as 100 for CIFAR-10, and 1000 for CIFAR-100 and TinyImageNet.

| iCaRL | A-GEM | EEIL | GD | DER++ | HAL |
|-------|-------|------|-----|-------|-------|
| 63.60 | 31.15 | 58.24 | 54.39 | 62.16 | 60.21 |

| DER | FOSTER | BEEF | MORE | ROW | TPL |
|-----|--------|------|------|-----|-----|
| 68.32 | 66.86 | 68.94 | 71.44 | 72.70 | **75.56** |

**More OOD methods.** To understand the effect of OOD detection on CIL, we applied 20 OOD detection methods to CIL and drew some interesting conclusions (see Appendix A). (1) There exists a linear relationship between OOD detection AUC and CIL ACC performances. (2) Different OOD detection methods result in similar TIL (task-incremental learning) ACC when applying HAT.

**More pre-trained models (visual encoders).** We also study TPL with different pre-trained models in Appendix D.5 (MAE, Dino, ViT and DeiT of different sizes). We found the pre-trained models based on supervised learning outperform self-supervised models in both CIL and TIL.

# 6 CONCLUSION

In this paper, we developed a novel approach for class incremental learning (CIL) via task-id prediction based on likelihood ratio. Recent studies (Kim et al., 2022a;b; 2023) suggested that OOD detection methods can be applied to perform task-id prediction in CIL and thus achieve the *state-of-the-art* performance. However, we argue that traditional OOD detection is not optimal for CIL as additional information in CIL can be leveraged to design a better and principled method for task-id prediction. Our experimental results show that our TPL outperforms strong baselines and has almost negligible catastrophic forgetting. Limitations of our approach are discussed in Appendix J.

## ACKNOWLEDGEMENTS

We sincerely thank Baizhou Huang of Peking University, Shanda Li of Carnegie Mellon University, and the anonymous reviewers of ICLR 2024 for providing valuable suggestions on this work.

## ETHICS STATEMENT

Since this research involves only classification learning using existing datasets downloaded from the public domain and our algorithms are not for any specific application but for solving the general problem of continual learning, we do not feel there are any possible ethical issues in this research.

## REPRODUCIBILITY STATEMENT

The source code of `TPL` has been public at https://github.com/linhaowei1/TPL. The proofs of Theorems 4.1 and 4.2 are provided in Appendix E. The training details and dataset details are given in Sec. 5.1 and Appendix I.

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

# Appendix of TPL

## Table of Contents

# A    A COMPREHENSIVE STUDY ON TIL+OOD BASED METHODS

Based on the previous research on CIL that combines TIL method and OOD detection (Kim et al., 2022b;a; 2023), we make a thorough study on this paradigm by benchmarking 20 popular OOD detection methods. This study provides a comprehensive understanding of these methods and draws some interesting conclusions, which we believe are beneficial to the future study of CIL.

## A.1    A UNIFIED CIL METHOD BASED ON TIL+OOD

We first describe how to design a CIL method based on TIL+OOD paradigm. As HAT is already a near-optimal solution for TIL (Kim et al., 2023) which has almost no CF, we choose HAT as the TIL method and only introduce variants in the OOD detection parts.

**Categorization of OOD detection methods.**  Usually, an OOD detection technique consists of two parts: (1) Training a classifier that can better distinguish IND data and OOD data in the feature / logits space; (2) Applying an inference-time OOD score to compute the INDness of the test case $x$. For training techniques, a group of methods such as LogitNorm Wei et al. (2022), MCDropout Gal & Ghahramani (2016), Mixup Thulasidasan et al. (2019), CutMix Yun et al. (2019) apply data augmentation or confidence regularization to improve OOD detection; another series of methods such as OE Hendrycks et al. (2018), MixOE Zhang et al. (2023), CMG Wang et al. (2022d), VOS Du et al. (2021) use auxiliary OOD data in training. For inference-time techniques, the OOD score can be computed based on *feature* (the hidden representations) or *logits* (the unnormalized softmax score). GradNorm Huang et al. (2021), KL-Matching Hendrycks et al. (2019), OpenMax Bendale & Boult (2016), ODIN Liang et al. (2017), MSP Hendrycks & Gimpel (2016), MLS Hendrycks et al. (2019), EBO Liu et al. (2020a), ReAct Sun et al. (2021) are logit-based methods, and Residual Ndiour et al. (2020), KNN Sun et al. (2022), MDS Lee et al. (2018b) are feature-based methods. Recently, VIM (Wang et al., 2022c) is based on both feature and logits. We study the aforementioned 20 OOD detection methods in this section. For simplicity, we use MSP (maximum softmax probability) as the OOD score for the 8 training-time techniques, and adopt standard supervised learning with cross-entropy loss for the inference-time OOD scores.

**A unified method for CIL based on TIL+OOD.** We introduce a unified CIL method that is based on TIL+OOD paradigm, which can be compatible with **any** OOD detection techniques. The details are as follows.

We first recall that HAT learns each task by performing two functions jointly *in training*: (1) learning a model for the task and (2) identifying the neurons that are important for the task and setting masks on them. When a new task is learned, the gradient flow through those masked neurons is blocked in the backward pass, which protects the models of the previous tasks to ensure no forgetting (CF). In the forward pass, no blocking is applied so that the tasks can share a lot of parameters or knowledge. Since (1) can be any supervised learning method, in our case, we replace it with an **OOD detection** method, which performs both in-distribution (IND) classification and out-of-distribution (OOD) detection.

*In testing*, we first predict the task-id to which the test instance $x$ belongs, and then perform the IND classification (within task prediction) in the task to obtain the predicted class. Let $S(x; t)$ be the OOD score of $x$ in task $t$ based on task $t$'s model, the task-id $\hat{t}$ can be predicted by identifying the task with the highest OOD score:

$$\hat{t} = \arg\max_t S(x; t)$$

We use argmax here as OOD score is defined to measure the IND-ness in the literature. Note that OOD score is produced by any OOD detection method.

## A.2    EXPERIMENTAL SETUP

**Backbone Architecture.** Following the main experiment in our paper, we use DeiT-S/16 (Touvron et al., 2021) that is pre-trained using 611 classes of ImageNet after removing 389 classes that are similar or identical to the classes of the experiment data CIFAR and TinyImageNet to prevent

Table 4: Average AUC and ACC results based on pre-trained DeiT on five datasets with five random seeds. Bold and underlined numbers indicate the best and second-best results, respectively. ♡: logit-based OOD scores, ♠: feature-based OOD scores, ◇: training-time techniques. The systems are divided into three categories by dashed lines. The first category includes post-hoc OOD detectors, the second category includes methods that exploit surrogate OOD data, and the last category includes methods that employ special training strategies.

| Method | C10-5T OOD | C10-5T CIL | C100-10T OOD | C100-10T CIL | C100-20T OOD | C100-20T CIL | T-5T OOD | T-5T CIL | T-10T OOD | T-10T CIL | Average OOD | Average CIL |
|---|---|---|---|---|---|---|---|---|---|---|---|---|
| GradNorm ♡ | $73.8^{\pm0.37}$ | $43.5^{\pm0.48}$ | $84.0^{\pm0.33}$ | $48.6^{\pm0.81}$ | $80.8^{\pm0.31}$ | $22.9^{\pm1.91}$ | $80.8^{\pm0.21}$ | $57.0^{\pm0.49}$ | $83.1^{\pm0.24}$ | $47.4^{\pm0.85}$ | 80.5 | 43.9 |
| KL-Matching ♡ | $75.2^{\pm0.31}$ | $48.8^{\pm0.44}$ | $84.1^{\pm0.35}$ | $48.7^{\pm0.74}$ | $82.6^{\pm0.32}$ | $25.6^{\pm1.81}$ | $77.8^{\pm0.25}$ | $47.6^{\pm0.58}$ | $73.6^{\pm0.29}$ | $31.6^{\pm0.79}$ | 80.7 | 40.5 |
| OpenMax ♡ | $94.1^{\pm0.14}$ | $84.1^{\pm0.29}$ | $84.7^{\pm0.26}$ | $50.9^{\pm0.62}$ | $90.6^{\pm0.24}$ | $57.9^{\pm0.47}$ | $74.1^{\pm0.19}$ | $48.1^{\pm0.24}$ | $71.7^{\pm0.09}$ | $32.0^{\pm0.33}$ | 83.0 | 54.6 |
| ODIN ♡ | $92.3^{\pm0.15}$ | $75.6^{\pm0.24}$ | $89.8^{\pm0.19}$ | $61.7^{\pm0.40}$ | $92.5^{\pm0.28}$ | $53.2^{\pm0.41}$ | $85.8^{\pm0.09}$ | $64.5^{\pm0.10}$ | $\underline{88.5}^{\pm0.07}$ | $59.5^{\pm0.25}$ | 86.4 | 62.9 |
| MSP ♡ | $93.5^{\pm0.03}$ | $82.4^{\pm0.12}$ | $89.0^{\pm0.11}$ | $62.9^{\pm0.24}$ | $91.7^{\pm0.25}$ | $59.5^{\pm0.49}$ | $81.5^{\pm0.10}$ | $59.2^{\pm0.21}$ | $84.4^{\pm0.09}$ | $54.0^{\pm0.21}$ | 88.0 | 63.6 |
| MLS ♡ | $94.0^{\pm0.11}$ | $\underline{84.9}^{\pm0.19}$ | $\underline{91.0}^{\pm0.18}$ | $\underline{69.2}^{\pm0.21}$ | $92.7^{\pm0.22}$ | $64.1^{\pm0.45}$ | $86.1^{\pm0.08}$ | $\underline{65.4}^{\pm0.21}$ | $\underline{88.5}^{\pm0.08}$ | $\underline{61.4}^{\pm0.27}$ | 90.4 | 69.0 |
| EBO ♡ | $94.0^{\pm0.11}$ | $\underline{84.9}^{\pm0.19}$ | $90.8^{\pm0.18}$ | $69.1^{\pm0.21}$ | $92.5^{\pm0.21}$ | $64.1^{\pm0.45}$ | $\underline{86.2}^{\pm0.08}$ | $\underline{65.4}^{\pm0.20}$ | $88.4^{\pm0.07}$ | $\underline{61.4}^{\pm0.28}$ | 90.4 | 69.0 |
| ReAct ♡ | $94.0^{\pm0.11}$ | $\underline{84.9}^{\pm0.20}$ | $90.8^{\pm0.19}$ | $69.1^{\pm0.22}$ | $92.5^{\pm0.19}$ | $64.1^{\pm0.39}$ | $\underline{86.2}^{\pm0.09}$ | $\underline{65.4}^{\pm0.21}$ | $88.4^{\pm0.05}$ | $\underline{61.4}^{\pm0.28}$ | 90.4 | 69.0 |
| KNN ♠ | $92.8^{\pm0.13}$ | $76.7^{\pm0.25}$ | $85.9^{\pm0.14}$ | $61.5^{\pm0.21}$ | $90.2^{\pm0.11}$ | $54.8^{\pm0.30}$ | $74.7^{\pm0.10}$ | $49.9^{\pm0.21}$ | $79.2^{\pm0.08}$ | $44.4^{\pm0.21}$ | 84.6 | 57.5 |
| Residual ♠ | $92.4^{\pm0.14}$ | $83.0^{\pm0.17}$ | $85.0^{\pm0.16}$ | $64.7^{\pm0.35}$ | $89.8^{\pm0.14}$ | $61.3^{\pm0.31}$ | $78.0^{\pm0.11}$ | $53.7^{\pm0.24}$ | $81.6^{\pm0.09}$ | $51.0^{\pm0.30}$ | 85.4 | 62.8 |
| MDS ♠ | $92.6^{\pm0.12}$ | $85.7^{\pm0.02}$ | $86.9^{\pm0.14}$ | $69.0^{\pm0.24}$ | $91.2^{\pm0.15}$ | $65.4^{\pm0.42}$ | $82.0^{\pm0.06}$ | $60.8^{\pm0.28}$ | $84.4^{\pm0.05}$ | $56.9^{\pm0.30}$ | 87.4 | 67.6 |
| VIM ♡♠ | $\mathbf{95.4}^{\pm0.07}$ | $\mathbf{89.0}^{\pm0.23}$ | $\underline{91.0}^{\pm0.12}$ | $\mathbf{72.8}^{\pm0.30}$ | $93.5^{\pm0.13}$ | $\mathbf{69.8}^{\pm0.55}$ | $\mathbf{86.3}^{\pm0.05}$ | $\mathbf{65.9}^{\pm0.23}$ | $\mathbf{88.7}^{\pm0.07}$ | $\mathbf{63.1}^{\pm0.42}$ | 91.0 | 72.1 |
| CMG ◇ | $92.3^{\pm0.21}$ | $80.7^{\pm0.40}$ | $85.2^{\pm0.28}$ | $56.2^{\pm0.60}$ | $90.1^{\pm0.24}$ | $53.1^{\pm0.81}$ | $80.5^{\pm0.12}$ | $56.9^{\pm0.25}$ | $83.1^{\pm0.10}$ | $50.5^{\pm0.31}$ | 86.2 | 59.5 |
| VOS ◇ | $93.1^{\pm0.10}$ | $82.1^{\pm0.31}$ | $86.7^{\pm0.22}$ | $59.4^{\pm0.58}$ | $90.9^{\pm0.18}$ | $56.0^{\pm0.66}$ | $82.4^{\pm0.10}$ | $59.7^{\pm0.21}$ | $83.8^{\pm0.10}$ | $51.8^{\pm0.34}$ | 87.8 | 61.8 |
| OE ◇ | $\underline{94.5}^{\pm0.10}$ | $84.3^{\pm0.24}$ | $90.9^{\pm0.15}$ | $66.7^{\pm0.31}$ | $\mathbf{94.0}^{\pm0.14}$ | $\underline{66.2}^{\pm0.40}$ | $82.4^{\pm0.08}$ | $60.5^{\pm0.19}$ | $85.7^{\pm0.10}$ | $56.4^{\pm0.33}$ | 89.5 | 66.8 |
| MixOE ◇ | $93.6^{\pm0.12}$ | $82.1^{\pm0.28}$ | $\mathbf{91.7}^{\pm0.19}$ | $67.6^{\pm0.37}$ | $\underline{93.9}^{\pm0.15}$ | $62.9^{\pm0.37}$ | $85.4^{\pm0.09}$ | $61.9^{\pm0.19}$ | $87.4^{\pm0.11}$ | $56.0^{\pm0.36}$ | 90.4 | 66.1 |
| LogitNorm ◇ | $93.1^{\pm0.11}$ | $82.2^{\pm0.21}$ | $89.0^{\pm0.17}$ | $64.3^{\pm0.35}$ | $91.9^{\pm0.14}$ | $59.7^{\pm0.31}$ | $81.7^{\pm0.05}$ | $58.7^{\pm0.15}$ | $84.6^{\pm0.06}$ | $53.4^{\pm0.37}$ | 88.1 | 63.7 |
| MCDropout ◇ | $92.4^{\pm0.17}$ | $80.7^{\pm0.27}$ | $87.8^{\pm0.19}$ | $61.7^{\pm0.38}$ | $91.3^{\pm0.16}$ | $57.5^{\pm0.44}$ | $80.9^{\pm0.08}$ | $57.5^{\pm0.19}$ | $84.0^{\pm0.08}$ | $52.2^{\pm0.32}$ | 87.3 | 65.5 |
| Mixup ◇ | $89.8^{\pm0.62}$ | $73.8^{\pm1.54}$ | $89.6^{\pm0.19}$ | $64.0^{\pm0.39}$ | $91.3^{\pm0.18}$ | $55.4^{\pm0.42}$ | $83.2^{\pm0.10}$ | $61.5^{\pm0.25}$ | $85.6^{\pm0.07}$ | $56.0^{\pm0.34}$ | 87.9 | 66.5 |
| CutMix ◇ | $90.4^{\pm0.26}$ | $69.0^{\pm0.71}$ | $89.5^{\pm0.16}$ | $62.4^{\pm0.34}$ | $90.9^{\pm0.15}$ | $50.6^{\pm0.40}$ | $82.6^{\pm0.12}$ | $61.3^{\pm0.25}$ | $85.1^{\pm0.22}$ | $55.2^{\pm0.51}$ | 87.7 | 59.7 |

information leak. We insert an adapter module (Houlsby et al., 2019) at each transformer layer. The adapter modules, classifiers and the layer norms are trained using `HAT` while the transformer parameters are fixed to prevent forgetting in the pre-trained network.

**Evaluation Protocol.** We compute AUC for OOD detection on each task model. The classes of the task are the IND classes while the classes of all other tasks of the dataset are the OOD classes. The evaluation metric for CIL is accuracy (ACC), which is measured after all tasks are learned. We report the average AUC value over all the tasks in each dataset, and the ACC of each dataset. Note that as `KNN` needs the training data at test time, in the CIL setting, we can only use the saved replay data of each task for its OOD score computation as the full data of previous tasks are not accessible in CIL. We also compute TIL accuracy in using difference OOD training-time techniques.

## A.3 RESULTS AND ANALYSIS

The experiment results are given in Table 4, which allow us to make some important observations.

**(1) OOD detection and CIL performances.** The OOD detection results in AUC here have similar trends as those in Yang et al. (2022) except `KNN`, which was considered as one of the best methods. But it is weak here because, as indicated above, in CIL, `KNN` can only use the replay data (which is very small) for each task to compute the OOD score. The CIL performances of the top OOD methods are competitive compared to CIL baselines in Table 1.

**(2) Similar TIL Performance.** We present the detailed results of TIL accuracy, for the OOD detection baselines as table 5. It is evident from the results that there is minimal variation in the performance of OOD detection methods across different datasets, with average values of 99.2±0.02, 95.7±0.12, 97.6±0.10, 84.3±0.86, 88.1±0.86 for C10-5T, C100-10T, C100-20T, T-5T, and T-10T, respectively. The observation that the majority of OOD methods exhibit negligible impact on the IND classification performance aligns with a previous benchmark study on OOD detection conducted by Yang et al. (2022). Notably, the aforementioned study investigated the finding on ResNet architecture (He et al., 2016) without pre-training, whereas our experiments involved a pre-trained DeiT model. This suggests that the finding extends to various model backbones.

**(3) Linear relationship between OOD AUC and CIL ACC.** We plot the relationship between OOD AUC and CIL performances in fig. 3. Interestingly, we see a linear relationship with Pearson correlation coefficients of 0.976, 0.811, 0.941, 0.963, and 0.980 for the 5 datasets, respectively. This finding suggests that improving OOD AUC can bring about a linear improvement of $\times 1.5 \sim 3.4$ (which is the slopes of the fitted linear function) on CIL ACC. Note that we are not conditioning

Table 5: Average IND classification ACC (TIL accuracy) results of pre-trained DeiT on five datasets with five random seeds. The baselines are divided into three categories as Table 1 in the main text. The first category includes post-hoc detectors, the second category includes methods that exploit surrogate OOD data, and the last category includes methods that employ special training strategies. ♡: logit-based OOD scores, ♠: feature-based OOD scores, ◇: training-time techniques. Note that the post-hoc methods only differ in the OOD score computation, which means they share the same trained model and thus have the same TIL ACC.

| | C10-5T | C100-10T | C100-20T | T-5T | T-10T | Average |
|---|---|---|---|---|---|---|
| GradNorm ♡ | | | | | | |
| KL-Matching ♡ | | | | | | |
| OpenMax ♡ | | | | | | |
| ODIN ♡ | | | | | | |
| MSP ♡ | | | | | | |
| MLS ♡ | 99.20 | 95.71 | 97.50 | 84.40 | 88.10 | 92.98 |
| EBO ♡ | | | | | | |
| ReAct ♡ | | | | | | |
| KNN ♠ | | | | | | |
| Residual ♠ | | | | | | |
| MDS ♠ | | | | | | |
| VIM ♡♠ | | | | | | |
| CMG ◇ | 99.20 | 95.70 | 97.61 | 84.52 | 88.30 | 93.07 |
| VOS ◇ | 99.20 | 95.66 | 97.59 | 84.51 | 88.30 | 93.05 |
| OE ◇ | 99.17 | 95.73 | 97.72 | 84.46 | 88.64 | 93.14 |
| MixOE ◇ | 99.21 | 95.82 | 97.80 | 82.06 | 86.53 | 92.28 |
| LogitNorm ◇ | 99.20 | 95.71 | 97.56 | 84.55 | 87.87 | 92.98 |
| MCDropout ◇ | 99.20 | 95.60 | 97.62 | 84.51 | 88.23 | 93.03 |
| Mixup ◇ | 99.24 | 95.57 | 97.71 | 85.04 | 88.47 | 93.21 |
| CutMix ◇ | 99.20 | 95.40 | 97.54 | 84.54 | 88.40 | 93.02 |

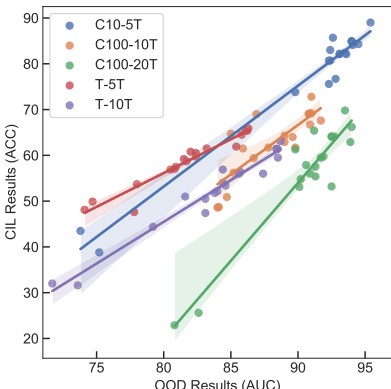

Figure 3: The correlation between OOD (AUC) and CIL (ACC) results. Each point denotes the AUC and ACC of one method in table 4 on the same dataset.

on TIL accuracy as (2) above showed the TIL results are similar for different training-time OOD techniques.

# B    OUTPUT CALIBRATION

We used the output calibration technique to balance the scales of task-id prediction scores for different tasks, which is motivated by Kim et al. (2022a;b). Even if the task-id prediction of each task-model is perfect, the system can make an incorrect task-id prediction if the magnitudes of the outputs across different tasks are different. As the task-specific modules are trained separately in HAT, it is useful to calibrate the outputs of different task modules.

To ensure that the output values are comparable, we calibrate the outputs by scaling $\sigma_1^{(t)}$ and shifting $\sigma_2^{(t)}$ parameters for each task. The optimal parameters $\{(\sigma_1^{(t)}, \sigma_2^{(t)})\}_{t=1}^{T} \in \mathbb{R}^{2T}$ ($T$ is the number of tasks) can be found by solving optimization problem using samples in the replay buffer *Buf*.

Specifically, we minimize the cross-entropy loss using SGD optimizer with batch size 64 for 100 epochs to find optimal calibration parameters $\{(\sigma_1^{(t)}, \sigma_2^{(t)})\}_{t=1}^{T}$:

$$\mathcal{L}_{calibration} = -\mathbb{E}_{(\boldsymbol{x},y) \in Buf} \log p(y|\boldsymbol{x}),$$

where $p(y|\boldsymbol{x})$ is computed using eq. (3) and calibration parameters:

$$p(y_j^{(t)}|\boldsymbol{x}) = \sigma_1^{(t)} \cdot \left[ softmax\left( f(h(\boldsymbol{x}; \phi^{(t)}); \theta^{(t)}) \right) \right]_j \cdot S(\boldsymbol{x}; t) + \sigma_2^{(t)}$$

Given the optimal parameters $\{(\tilde{\sigma}_1^{(t)}, \tilde{\sigma}_2^{(t)})\}_{t=1}^{T}$, we make the final prediction as:

$$\hat{y} = \underset{1 \leq t \leq T, 1 \leq j \leq |\mathcal{Y}_t|}{\arg\max} p(y_j^{(t)}|\boldsymbol{x})$$

## C    EVALUATION METRICS

### C.1    DEFINITIONS OF AIA. AND LAST ACCURACY

Fllowing (Kim et al., 2022a), we give the formal definitions of *average incremental accuracy* (**AIA** in Table 1 in the main text, denoted as $A_{AIA}$) and *accuracy after learning the final task* (**Last** in Table 1 in the main text, denoted as $A_{last}$). Let the accuracy after learning the task $t$ be:

$$A^{(\leq t)} = \frac{\#\textit{correctly classified samples in } \bigcup_{k=1}^{t} \mathcal{D}_{test}^{(k)}}{\#\textit{samples in } \bigcup_{k=1}^{t} \mathcal{D}_{test}^{(k)}}$$

Let $T$ be the last task. Then, $A_{last} = A^{(\leq T)}$ and $A_{AIA} = \frac{1}{T} \sum_{k=1}^{T} A^{(\leq k)}$.

Here $\mathcal{D}_{test}^{(k)}$ denotes the test-set for task $k$, and $\#$ denotes "the number of". To put it simply, $A^{(\leq t)}$ means the accuracy of all the test data from task 1 to task $t$.

### C.2    RECTIFIED FORGETTING RATE FOR CIL

Apart from the *classification accuracy* (ACC), we report another popular CIL evaluation metric *average forgetting rate*. The popular definition of average forgetting rate is the following

$$\mathcal{F}^{(t)} = \frac{1}{t-1} \sum_{i=1}^{t-1} (A_i^{(i)} - A_i^{(t)}),$$

where $A_i^{(t)}$ is the accuracy of task $i$'s test set on the CL model after task $t$ is learned (Liu et al., 2020b), which is also referred to as *backward transfer* in other literature (Lopez-Paz & Ranzato, 2017).

However, this formula is only suitable for TIL but not appropriate for CIL. As the task-id for each test sample is given in testing for TIL, all the test samples from task $i$ will be classified into one of the classes of task $i$. If there is no forgetting for a TIL model, then $A_i^{(t)}$ will be equal to $A_i^{(i)}(i < t)$ in such a within-task classification, where the number of classes is fixed. But in CIL, the task-id is not provided in testing and we are not doing within-task classification. **Even if in the Non-CL setting as more tasks or classes are learned, the classification accuracy will usually decrease for the same test set due to the nature of multi-class classification.** That is, $A_i^{(t)} < A_i^{(i)}(i < t)$ is usually true as at task $i$ there are fewer learned classes than at task $t$. Their difference is not due to forgetting.

Furthermore, as we discussed in Section 1, forgetting (CF) is not the only issue of CIL. Inter-class separation (ICS) is another important one. When considering the performance degradation of each task during continual learning, it is hard to disentangle the effects of these two factors. Our rectified average forgetting rate metric for CIL considers both forgetting and ICS. Two new average forgetting rates for CIL are defined, one for the case where we use the Last accuracy as the evaluation metric and one for the case where we use AIA as the evaluation metric:

$$\mathcal{F}_{CIL, Last}^{(t)} = \frac{1}{t} \sum_{i=1}^{t} (A_i^{(t,NCL)} - A_i^{(t)})$$

$$\mathcal{F}_{CIL, AIA}^{(t)} = \frac{1}{t} \sum_{i=1}^{t} \mathcal{F}_{CIL, Last}^{(i)},$$

where NCL means Non-CL, $A_i^{(t)}$ is the accuracy of task $i$'s test set on the CL model after task $t$ is learned, and $A_i^{(t,NCL)}$ is the accuracy of task $i$'s test set on the Non-CL model that learns all tasks from 1 to $t$. If the test dataset sizes are the same across different tasks, then $\mathcal{F}_{CIL, Last}^{(t)}$ is equal to $A_{last, NCL} - A_{last}$, where $A_{last}$ is defined in appendix C.1 and $A_{last, NCL}$ is the $A_{last}$ of Non-CL. The intuition of introducing NCL performance is to address the loss of accuracy for each task when more tasks are learned.

Table 6: Forgetting rate (%) for CIL on the five datasets of the baselines in Table 1. The lower the rate, the better the method is.

| Method | C10-5T $\mathcal{F}^{(T)}_{CIL, Last}$ | C10-5T $\mathcal{F}^{(T)}_{CIL, AIA}$ | C100-10T $\mathcal{F}^{(T)}_{CIL, Last}$ | C100-10T $\mathcal{F}^{(T)}_{CIL, AIA}$ | C100-20T $\mathcal{F}^{(T)}_{CIL, Last}$ | C100-20T $\mathcal{F}^{(T)}_{CIL, AIA}$ | T-5T $\mathcal{F}^{(T)}_{CIL, Last}$ | T-5T $\mathcal{F}^{(T)}_{CIL, AIA}$ | T-10T $\mathcal{F}^{(T)}_{CIL, Last}$ | T-10T $\mathcal{F}^{(T)}_{CIL, AIA}$ | Average $\mathcal{F}^{(T)}_{CIL, Last}$ | Average $\mathcal{F}^{(T)}_{CIL, AIA}$ |
|---|---|---|---|---|---|---|---|---|---|---|---|---|
| OWM | 54.10 | 41.01 | 61.37 | 47.10 | 65.78 | 54.95 | 47.97 | 31.85 | 55.00 | 41.28 | 56.84 | 43.24 |
| ADAM | 11.87 | 6.68 | 21.55 | 14.65 | 23.77 | 16.64 | 22.41 | 15.18 | 22.84 | 15.59 | 20.49 | 13.75 |
| PASS | 9.58 | 7.98 | 13.86 | 10.19 | 15.99 | 11.11 | 11.49 | 9.91 | 14.18 | 9.70 | 13.02 | 9.78 |
| HAT$_{CIL}$ | 13.39 | 5.95 | 19.85 | 13.21 | 23.22 | 18.41 | 13.30 | 7.65 | 18.49 | 11.40 | 17.65 | 11.32 |
| iCaRL | 39.46 | 28.82 | 57.55 | 43.37 | 60.77 | 51.56 | 41.99 | 27.77 | 50.62 | 37.45 | 50.08 | 37.79 |
| A-GEM | 8.24 | 7.27 | 13.86 | 10.70 | 13.61 | 10.47 | 19.39 | 15.67 | 20.64 | 13.47 | 15.15 | 11.52 |
| EEIL | 13.45 | 6.51 | 14.68 | 6.10 | 18.97 | 7.99 | 19.18 | 10.40 | 22.14 | 10.49 | 17.68 | 8.30 |
| GD | 6.63 | 2.79 | 18.40 | 6.69 | 22.66 | 9.10 | 19.51 | 9.52 | 30.04 | 13.12 | 19.45 | 8.24 |
| DER++ | 9.00 | 4.18 | 9.46 | 4.31 | 10.76 | 4.74 | 12.95 | 6.71 | 15.34 | 6.82 | 11.50 | 5.35 |
| HAL | 11.41 | 10.01 | 15.59 | 9.78 | 15.39 | 9.68 | 19.72 | 11.72 | 17.27 | 12.55 | 15.88 | 10.75 |
| DER++ | 11.16 | 8.00 | 13.03 | 6.56 | 12.73 | 5.81 | 16.68 | 10.48 | 18.32 | 9.89 | 14.38 | 8.15 |
| FOSTER | 9.70 | 5.47 | 11.07 | 6.04 | 9.85 | 4.51 | 18.08 | 7.08 | 16.82 | 7.03 | 13.10 | 6.03 |
| BEEF | 8.69 | 3.91 | 10.67 | 5.29 | 10.88 | 6.08 | 11.11 | 5.82 | 14.36 | 5.87 | 11.14 | 5.39 |
| MORE | 6.63 | 2.78 | 12.5 | 5.96 | 12.2 | 5.94 | 7.55 | 3.00 | 9.46 | 4.29 | 9.68 | 4.39 |
| ROW | 4.82 | 2.56 | 8.04 | 4.33 | 8.16 | 4.41 | 7.41 | 2.87 | 9.31 | 4.12 | 7.55 | 3.66 |
| TPL | 3.46 | 1.90 | 6.23 | 3.10 | 6.42 | 3.07 | 3.88 | 0.26 | 5.32 | 1.31 | 5.06 | 1.93 |
| TPL$_{PFI}$ | 2.04 | 1.07 | 1.18 | 1.44 | 2.75 | 1.57 | 1.49 | 1.07 | 1.84 | 1.26 | 1.86 | 1.28 |

Table 6 shows the *average forgetting rates* of each system based on the new definitions. We clearly observe that TPL and TPL$_{PFI}$ have the lowest average forgetting rates on the five datasets among all systems. SLDA and L2P are not included as they use different architectures and cannot take the Non-CL results in Table 1 in the main text as the upper bounds or NCL results needed in the proposed formulas above.

It is important to note that for both TPL and TPL$_{PFI}$, the forgetting rate mainly reflects the performance loss due to the ICS problem rather than the traditional *catastrophic forgetting* (CF) caused by network parameter interference in the incremental learning of different tasks because the TIL method HAT has effectively eliminated CF in TPL and TPL$_{PFI}$.

# D    ADDITIONAL EXPERIMENTAL RESULTS

## D.1    CIL EXPERIMENTS ON A LARGER DATASET

Table 7: The CIL ACC after the final task on ImageNet380-10T. We highlight the best results in bold.

| $\text{HAT}_{CIL}$ | ADAM | SLDA | PASS | L2P | iCaRL | A-GEM | EEIL |
|---|---|---|---|---|---|---|---|
| $71.20^{\pm0.99}$ | $62.10^{\pm0.91}$ | $65.78^{\pm0.05}$ | $65.27^{\pm1.24}$ | $47.89^{\pm3.24}$ | $62.23^{\pm0.66}$ | $30.38^{\pm10.02}$ | $63.37^{\pm0.49}$ |

| DER++ | HAL | DER | FORSTER | BEEF | MORE | ROW | TPL |
|---|---|---|---|---|---|---|---|
| $66.53^{\pm2.36}$ | $64.83^{\pm2.60}$ | $69.19^{\pm1.36}$ | $68.07^{\pm1.88}$ | $70.07^{\pm1.41}$ | $72.10^{\pm1.44}$ | $74.52^{\pm1.38}$ | $\mathbf{78.49}^{\pm0.89}$ |

To assess the performance of our proposed `TPL` on large-scale datasets, we use ImageNet-1k (Russakovsky et al., 2015), a widely recognized benchmark dataset frequently examined in the CIL literature. However, due to the nature of our experiments, which involve a DeiT backbone pretrained on 611 ImageNet classes after excluding 389 classes similar to those in CIFAR and TinyImageNet, we cannot directly evaluate our model on the original ImageNet dataset to avoid potential information leak.

To overcome this limitation, we created a new benchmark dataset called **ImageNet-380**. We randomly selected 380 classes from the remaining 389 classes, excluding those similar to CIFAR and TinyImageNet, from the original set of 1k classes in the full ImageNet dataset. This new dataset consists of approximately 1,300 color images per class. For ImageNet-380, we divided the classes into 10 tasks, with each task comprising 38 classes. We set the replay buffer size to 7600, with 20 samples per class, which is a commonly used number in replay-based methods. We refer to these experiments as ImageNet380-10T. For other training configurations, we kept them consistent with the experiments conducted on T-10T.

The CIL Last ACC achieved after the final task on ImageNet380-10T can be found in Table 7. Notably, the results obtained by `TPL` still exhibit a significant improvement over the baselines, with a 3.97% higher ACC compared to the best baseline method `ROW`. The results further provide strong evidence supporting the effectiveness of our proposed `TPL` system.

## D.2 CIL Experiments without Pre-training

Table 8: CIL accuracy (%) after the final task (last) based on ResNet-18 without pre-training over 5 runs with random seeds. "-XT": X number of tasks. The best result in each column is highlighted in bold.

| | C10-5T | C100-10T | C100-20T | T-5T | T-10T | Average |
|---|---|---|---|---|---|---|
| OWM | $51.8^{\pm0.05}$ | $28.9^{\pm0.60}$ | $24.1^{\pm0.26}$ | $10.0^{\pm0.55}$ | $8.6^{\pm0.42}$ | 24.7 |
| PASS | $47.3^{\pm0.98}$ | $33.0^{\pm0.58}$ | $25.0^{\pm0.69}$ | $28.4^{\pm0.51}$ | $19.1^{\pm0.46}$ | 30.6 |
| EEIL | $64.5^{\pm0.93}$ | $52.3^{\pm0.83}$ | $48.0^{\pm0.44}$ | $38.2^{\pm0.54}$ | $28.7^{\pm0.87}$ | 46.3 |
| GD | $65.5^{\pm0.94}$ | $51.4^{\pm0.83}$ | $50.3^{\pm0.88}$ | $38.9^{\pm1.01}$ | $29.5^{\pm0.68}$ | 47.1 |
| HAL | $63.7^{\pm0.91}$ | $51.3^{\pm1.22}$ | $48.5^{\pm0.71}$ | $38.1^{\pm0.97}$ | $30.3^{\pm1.05}$ | 46.4 |
| A-GEM | $64.6^{\pm0.72}$ | $50.5^{\pm0.73}$ | $47.3^{\pm0.87}$ | $37.3^{\pm0.89}$ | $29.4^{\pm0.95}$ | 45.8 |
| HAT$_{CIL}$ | $62.7^{\pm1.45}$ | $41.1^{\pm0.93}$ | $25.6^{\pm0.51}$ | $38.5^{\pm1.85}$ | $29.8^{\pm0.65}$ | 39.5 |
| iCaRL | $63.4^{\pm1.11}$ | $51.4^{\pm0.99}$ | $47.8^{\pm0.48}$ | $37.0^{\pm0.41}$ | $28.3^{\pm0.18}$ | 45.6 |
| DER++ | $66.0^{\pm1.20}$ | $53.7^{\pm1.20}$ | $46.6^{\pm1.44}$ | $35.8^{\pm0.77}$ | $30.5^{\pm0.47}$ | 46.5 |
| DER | $62.1^{\pm0.97}$ | $\mathbf{64.5}^{\pm0.85}$ | $\mathbf{62.5}^{\pm0.76}$ | $43.6^{\pm0.77}$ | $38.3^{\pm0.82}$ | 54.2 |
| FOSTER | $65.4^{\pm1.05}$ | $62.5^{\pm0.84}$ | $56.3^{\pm0.71}$ | $40.5^{\pm0.92}$ | $36.4^{\pm0.85}$ | 52.2 |
| BEEF | $67.3^{\pm1.07}$ | $60.9^{\pm0.87}$ | $56.7^{\pm0.72}$ | $44.1^{\pm0.85}$ | $37.9^{\pm0.95}$ | 53.4 |
| MORE | $70.6^{\pm0.74}$ | $57.5^{\pm0.68}$ | $51.3^{\pm0.89}$ | $41.2^{\pm0.81}$ | $35.4^{\pm0.72}$ | 51.2 |
| ROW | $74.6^{\pm0.89}$ | $58.2^{\pm0.67}$ | $52.1^{\pm0.91}$ | $42.3^{\pm0.69}$ | $38.2^{\pm1.34}$ | 53.1 |
| TPL | $\mathbf{78.4}^{\pm0.78}$ | $62.2^{\pm0.52}$ | $55.8^{\pm0.57}$ | $\mathbf{48.2}^{\pm0.64}$ | $\mathbf{42.9}^{\pm0.45}$ | $\mathbf{57.5}$ |

The experimental setup and results for CIL baselines without pre-training are presented in this section.

**Training details.** We follow Kim et al. (2022b) to use ResNet-18 (He et al., 2016) for all the datasets (CIFAR-10, CIFAR-100, TinyImageNet) and all the baselines excluding OWM. MORE, ROW, and our TPL are designed for pre-trained models, and we adapt them by applying HAT on ResNet-18. All other baselines adopted ResNet-18 in their original paper. OWM adopts AlexNet as it is hard to apply the method to the ResNet. For the replay-based methods, we also use the same buffer size as specified in Section 5.1. We use the hyper-parameters suggested by their original papers. For MORE, ROW, and our proposed TPL, we follow the hyper-parameters used in HAT for training.

**Experimental Results.** The results for CIL Last accuracy (ACC) after the final task are shown in Table 8. We can observe that the network-expansion-based approaches (DER, FOSTER, BEEF) and approaches that predict task-id based on TIL+OOD (MORE, ROW) are two competitive groups of CIL baselines. Our proposed TPL achieves the best performance on C10-5T, T-5T, and T-10T, while DER achieves the best on C100-10T and C100-20T. Overall, our proposed TPL achieves the best average accuracy over the 5 datasets (57.5%) while DER ranks the second (with an average ACC of 54.2%).

## D.3   CIL Experiments on Smaller Replay Buffer Size

Table 9: CIL accuracy (%) after the final task (Last) with smaller replay buffer size over 5 runs with random seeds. "-XT": X number of tasks. The best result in each column is highlighted in bold. The replay buffer size is set to 100 for CIFAR-10, and 1000 for CIFAR-100 and TinyImageNet. The pre-trained model is used.

| | C10-5T | C100-10T | C100-20T | T-5T | T-10T | Average |
|---|---|---|---|---|---|---|
| iCaRL | $86.08^{\pm1.19}$ | $66.96^{\pm2.08}$ | $68.16^{\pm0.71}$ | $47.27^{\pm3.22}$ | $49.51^{\pm1.87}$ | 63.60 |
| A-GEM | $56.64^{\pm4.29}$ | $23.18^{\pm2.54}$ | $20.76^{\pm2.88}$ | $31.44^{\pm3.84}$ | $23.73^{\pm6.27}$ | 31.15 |
| EEIL | $77.44^{\pm3.04}$ | $62.95^{\pm0.68}$ | $57.86^{\pm0.74}$ | $48.36^{\pm1.38}$ | $44.59^{\pm1.72}$ | 58.24 |
| GD | $85.96^{\pm1.64}$ | $57.17^{\pm1.06}$ | $50.30^{\pm0.58}$ | $46.09^{\pm1.77}$ | $32.41^{\pm2.75}$ | 54.39 |
| DER++ | $80.09^{\pm3.00}$ | $64.89^{\pm2.48}$ | $65.84^{\pm1.46}$ | $50.74^{\pm2.41}$ | $49.24^{\pm5.01}$ | 62.16 |
| HAL | $79.16^{\pm4.56}$ | $62.65^{\pm0.83}$ | $63.96^{\pm1.49}$ | $48.17^{\pm2.94}$ | $47.11^{\pm6.00}$ | 60.21 |
| DER | $85.11^{\pm1.44}$ | $72.31^{\pm0.78}$ | $70.25^{\pm0.98}$ | $58.07^{\pm1.40}$ | $55.85^{\pm1.23}$ | 68.32 |
| FOSTER | $84.99^{\pm0.89}$ | $70.25^{\pm0.58}$ | $71.14^{\pm0.76}$ | $53.35^{\pm0.54}$ | $54.58^{\pm0.85}$ | 66.86 |
| BEEF | $86.20^{\pm1.59}$ | $70.87^{\pm2.77}$ | $70.44^{\pm1.24}$ | $60.15^{\pm0.98}$ | $57.02^{\pm0.87}$ | 68.94 |
| MORE | $88.13^{\pm1.16}$ | $71.69^{\pm0.11}$ | $71.29^{\pm0.55}$ | $64.17^{\pm0.77}$ | $61.90^{\pm0.90}$ | 71.44 |
| ROW | $89.70^{\pm1.54}$ | $73.63^{\pm0.12}$ | $71.86^{\pm0.07}$ | $65.42^{\pm0.55}$ | $62.87^{\pm0.53}$ | 72.70 |
| TPL | $\mathbf{91.76}^{\pm0.44}$ | $\mathbf{75.83}^{\pm0.28}$ | $\mathbf{75.65}^{\pm0.54}$ | $\mathbf{68.08}^{\pm0.61}$ | $\mathbf{66.48}^{\pm0.47}$ | **75.56** |

## D.4  ABLATION ON $\beta_1$ AND $\beta_2$

Table 10: The CIL accuracy (%) after leraning the final task (Last) of our TPL on C10-5T for different $\beta_1$ and $\beta_2$.

| $\beta_1$ \ $\beta_2$ | 15.0 | 17.5 | 20.0 | 22.5 | 25.0 |
|---|---|---|---|---|---|
| 0.5 | 91.9 | 91.9 | 91.8 | 92.5 | 92.2 |
| 0.6 | 92.1 | 91.0 | 92.2 | 91.6 | 92.0 |
| 0.7 | 91.8 | 92.1 | 92.3 | 92.4 | 91.9 |
| 0.8 | 92.0 | 92.4 | 91.7 | 91.9 | 91.7 |
| 0.9 | 92.3 | 92.5 | 92.0 | 91.9 | 91.5 |

$\beta_1$ and $\beta_2$ are two scaling hyper-parameters used in the definition of task-id prediction score $S_{TPL}^{(t)}(\boldsymbol{x})$. Table 10 shows an ablation for different $\beta_1$ and $\beta_2$ on C10-5T, which indicates that $\beta_1$ and $\beta_2$ do not affect results much. For other hyper-parameters used in TPL, see Appendix I.1 for more details.

## D.5 EXPERIMENTS BASED ON MORE PRE-TRAINED MODELS

Table 11: **Last** TIL and CIL accuracy results (after the last task is learned) for TPL based on different pre-trained visual encoders or models.

| Visual Encoder | Pre-training | C10-5T TIL | C10-5T CIL | C100-10T TIL | C100-10T CIL | C100-20T TIL | C100-20T CIL | T-5T TIL | T-5T CIL | T-10T TIL | T-10T CIL | Average TIL | Average CIL |
|---|---|---|---|---|---|---|---|---|---|---|---|---|---|
| DeiT-small-IN661 (TPL) | supervised | 99.20 | 92.33 | 95.71 | 76.53 | 97.50 | 76.34 | 84.40 | 68.64 | 88.10 | 67.20 | 92.98 | 76.21 |
| ViT-tiny | | 98.80 | 91.38 | 95.36 | 76.79 | 97.52 | 75.83 | 82.37 | 71.85 | 85.10 | 70.18 | 91.83 | 77.21 |
| DeiT-tiny | supervised | 98.85 | 90.79 | 94.80 | 74.01 | 97.67 | 73.21 | 83.58 | 72.46 | 86.54 | 71.71 | 92.29 | 76.44 |
| ViT-small | | 99.43 | 95.57 | 97.51 | 84.52 | 98.76 | 83.94 | 89.49 | 81.80 | 91.30 | 80.94 | 95.30 | 85.35 |
| DeiT-small (TPL$_{PFI}$) | | 99.24 | 94.86 | 96.79 | 82.43 | 97.78 | 80.86 | 89.78 | 84.06 | 92.51 | 83.87 | 95.22 | 85.22 |
| ViT-small-Dino | self-supervised | 98.75 | 87.82 | 94.51 | 73.83 | 96.95 | 72.62 | 80.77 | 68.78 | 82.49 | 65.97 | 90.69 | 73.80 |
| ViT-base-MAE | self-supervised | 99.09 | 88.82 | 93.42 | 67.47 | 96.77 | 69.52 | 79.58 | 65.94 | 81.76 | 63.10 | 90.12 | 70.97 |

In this section, we conduct an ablation study on the visual encoder (pre-trained model/network). To prevent data contamination or information leak, TPL uses DeiT-S/16 pre-trained with 611 classes of ImageNet (DeiT-small-IN661) after removing 389 classes that overlap with classes in the continual learning datasets. Here we dismiss this limitation and experiment on more open-sourced pre-trained visual encoders trained using the full ImageNet. The results are reported in Table 11, which also includes the TIL accuracy results if the task-id is provided for each test instance during testing. We show DeiT-small-IN661, which is our TPL, in the first row of the table. It is the backbone used in our main experiments (Table 1) (IN: ImageNet). The other visual encoders are open-sourced in the timm (Wightman, 2019) library. We note that vanilla ViT and DeiT are pre-trained on ImageNet using **supervised training**, thereby there exists an information leak for CIFAR and Tiny-ImageNet datasets used in continual learning. The details of the models are as follows:

- ViT-tiny: The full model name in timm is "vit_tiny_patch16_224". It is trained on ImageNet (with additional augmentation and regularization) using supervised learning.

- DeiT-tiny: The full model name in timm is "deit_tiny_patch16_224". It is trained on ImageNet (with additional augmentation and regularization) using supervised learning and distillation.

- ViT-small: The small version of ViT ("vit_small_patch16_224").

- DeiT-small: The small version of DeiT ("deit_small_patch16_224").

- ViT-small-Dino: The small version of ViT trained with self-supervised DINO method (Caron et al., 2021).

- ViT-base-MAE: The base version of ViT trained with self-supervised MAE method (He et al., 2021).

**Analysis.** We found that different pre-trained visual encoders have varied CIL results. Compared to DeiT-small-IN661, the pre-trained small ViT and DeiT that use the full ImageNet to conduct supervised learning have overall better performance, which is not surprising due to the class overlap as we discussed above. For the self-supervised visual encoders (Dino, MAE), the performances are worse than the supervised pre-trained visual encoders. We hypothesize that the variance between different visual encoders is mainly rooted in the learned feature representations during pre-training. It is an interesting future work to design an optimal pre-training strategy for continual learning that does not need supervised data.

# E    Theoretical Justifications

## E.1    Preliminary

In this section, we first give some primary definitions and notations of *statistical hypothesis testing*, *rejection region* and *uniformly most powerful (UMP) test*.

**Definition 1 (statistical hypothesis testing and rejection region)** *Consider testing a null hypothesis $H_0 : \theta \in \Theta_0$ against an alternative hypothesis $H_1 : \theta \in \Theta_1$, where $\Theta_0$ and $\Theta_1$ are subsets of the parameter space $\Theta$ and $\Theta_0 \cap \Theta_1 = \emptyset$. A test consists of a test statistic $T(X)$, which is a function of the data $\boldsymbol{x}$, and a rejection region $\mathcal{R}$, which is a subset of the range of $T$. If the observed value $t$ of $T$ falls in $\mathcal{R}$, we reject $H_0$.*

**Type I error** occurs when we reject a true null hypothesis $\mathcal{H}_0$. The probability of making Type I error is usually denoted by $\alpha$. **Type II error** occurs when we *fail to reject* a false null hypothesis $\mathcal{H}_0$. The **level of significance** $\alpha$ is the probability we are willing to risk rejecting $\mathcal{H}_0$ when it is true. Typically $\alpha = 0.1, 0.05, 0.01$ are used.

**Definition 2 (UMP test)** *Denote the power function $\beta_{\mathcal{R}}(\theta) = P_\theta(T(x) \in \mathcal{R})$, where $P_\theta$ denotes the probability measure when $\theta$ is the true parameter. A test with a test statistic $T$ and rejection region $\mathcal{R}$ is called a **uniformly most powerful (UMP) test** at significance level $\alpha$ if it satisfies two conditions:*

1. *$\sup_{\theta \in \Theta_0} \beta_{\mathcal{R}}(\theta) \leq \alpha$.*

2. *$\forall \theta \in \Theta_1, \beta_{\mathcal{R}}(\theta) \geq \beta_{R'}(\theta)$ for every other test $t'$ with rejection region $R'$ satisfying the first condition.*

From the definition we see, the UMP test ensures that the probability of Type I error is less than $\alpha$ (with the first condition), while achieves the lowest Type II error (with the second condition). Therefore, UMP is considered an optimal solution in statistical hypothesis testing.

## E.2    Proof of Theorem 4.1

**Lemma E.1** *(Neyman & Pearson, 1933) Let $\{X_1, X_2, ..., X_n\}$ be a random sample with likelihood function $L(\theta)$. The UMP test of the simple hypothesis $H_0 : \theta = \theta_0$ against the simple hypothetis $H_a : \theta = \theta_a$ at level $\alpha$ has a rejection region of the form:*

$$\frac{L(\theta_0)}{L(\theta_a)} < k$$

*where $k$ is chosen so that the probability of a type I error is $\alpha$.*

Now the proof of Theorem 4.1 is straightforward. From Lemma E.1, the UMP test for Equation (4) in the main text has a rejection region of the form:

$$\frac{p_t(\boldsymbol{x})}{p_{t^c}(\boldsymbol{x})} < \lambda_0$$

where $\lambda_0$ is chosen so that the probability of a type I error is $\alpha$.

## E.3    Proof of Theorem 4.2

Note that the AUC is computed as the *area under the ROC curve*. A ROC curve shows the trade-off between true positive rate (TPR) and false positive rate (FPR) across different decision thresholds. Therefore,

$$AUC = \int_0^1 (TPR) \, \mathrm{d}(FPR) \tag{12}$$

$$= \int_0^1 (1 - FPR) \, \mathrm{d}(TPR) \tag{13}$$

$$= \int_0^1 \beta_{\mathcal{R}}(\theta_{t^c}) \, \mathrm{d}(1 - \beta_{\mathcal{R}}(\theta_t)) \tag{14}$$

$$= \int_0^1 \beta_{\mathcal{R}}(\theta_{t^c}) \, \mathrm{d}\beta_{\mathcal{R}}(\theta_t) \tag{15}$$

where FPR and TPR are *false positive rate* and *true positive rate*. Therefore, an optimal AUC requires UMP test of any given level $\alpha = \beta_{\mathcal{R}}(\theta_t)$ except on a null set.

### E.4 DISTANCE-BASED OOD DETECTORS ARE IND DENSITY ESTIMATORS

In this section, we show that $S_{MD}(\boldsymbol{x})$ (MD: *Mahalanobis distance*) and $S_{KNN}(\boldsymbol{x})$ (KNN: *k-nearest neighbor*) defined in Equation (16) and Equation (17) are IND (in-distribution) density estimators under different assumptions (We omit the superscript $(t)$ for simplicity):

$$S_{MD}(\boldsymbol{x}) = 1 / \min_{c \in \mathcal{Y}} (\boldsymbol{z} - \boldsymbol{\mu}_c)^T \boldsymbol{\Sigma}^{-1} (\boldsymbol{z} - \boldsymbol{\mu}_c), \tag{16}$$

$$S_{KNN}(\boldsymbol{x}; \mathcal{D}) = -||\boldsymbol{z}^* - kNN(\boldsymbol{z}^*; \mathcal{D}^*)||_2. \tag{17}$$

In Equation (16), $\boldsymbol{\mu}_c$ is the class centroid for class $c$ and $\Sigma$ is the global covariance matrix, which are estimated on IND training corpus $\mathcal{D}$. In Equation (17), $|| \cdot ||_2$ is Euclidean norm, $\boldsymbol{z}^* = \boldsymbol{z}/||\boldsymbol{z}||_2$ denotes the normalized feature $\boldsymbol{z}$, and $\mathcal{D}^*$ denotes the set of normalized features from training set $\mathcal{D}$. $kNN(\boldsymbol{z}^*; \mathcal{D}^*)$ denotes the $k$-nearest neighbor of $\boldsymbol{z}^*$ in set $\mathcal{D}^*$.

Assume we have a feature encoder $\phi : \mathcal{X} \to \mathbb{R}^m$, and in training time we empirically observe $n$ IND samples $\{\phi(\boldsymbol{x}_1), \phi(\boldsymbol{x}_2)...\phi(\boldsymbol{x}_n)\}$.

**Analysis of the MD score.** Denote $\Sigma$ to be the covariance matrix of $\phi(\boldsymbol{x})$. The final feature we extract from data $\boldsymbol{x}$ is:

$$\boldsymbol{z}(\boldsymbol{x}) = A^{-1}\phi(\boldsymbol{x})$$

where $AA^T = \Sigma$. Note that the covariance of $\boldsymbol{z}$ is $\mathcal{I}$.

Given a class label $c$, we assume the distribution $z(x|c)$ follows a Gaussian $\mathcal{N}(A^{-1}\mu_c, \mathcal{I})$. Immediately we have $\boldsymbol{\mu}_c$ to be the class centroid for class $c$ under the maximum likelihood estimation. We can now clearly address the relation between MD score and IND density ($p(\boldsymbol{x})$):

$$S_{MD}(\boldsymbol{x}) = 1/(-2 \max_{c \in \mathcal{Y}} (\ln p(\boldsymbol{x}|c)) - m \ln 2\pi)$$

**Analysis of KNN score.** The normalized feature $\boldsymbol{z}(x) = \phi(\boldsymbol{x})/||\phi(\boldsymbol{x})||_2$ is used for OOD detection. The probability density of $\boldsymbol{z}$ can be attained by:

$$p(\boldsymbol{z}) = \lim_{r \to 0} \frac{p(\boldsymbol{z}' \in B(\boldsymbol{z}, r))}{|B(z, r)|}$$

where $B(\boldsymbol{z}, r) = \{\boldsymbol{z}' : ||\boldsymbol{z}' - \boldsymbol{z}||_2 \leq r \wedge ||\boldsymbol{z}'|| = 1\}$

Assuming each sample $\boldsymbol{z}(\boldsymbol{x}_i)$ is $i.i.d$ with a probability mass $1/n$, the density can be estimated by KNN distance. Specifically, $r = ||\boldsymbol{z} - kNN(\boldsymbol{z})||_2$, $p(\boldsymbol{z}' \in B(\boldsymbol{z}, r)) = k/n$ and $|B(\boldsymbol{z}, r)| = \frac{\pi^{(m-1)/2}}{\Gamma(\frac{m-1}{2} + 1)} r^{m-1} + o(r^{m-1})$, where $\Gamma$ is Euler's gamma function. When $n$ is large and $k/n$ is small, we have the following equations:

$$p(\boldsymbol{x}) \approx \frac{k\Gamma(\frac{m-1}{2} + 1)}{\pi^{(m-1)/2} n r^{m-1}}$$

$$S_{KNN}(\boldsymbol{x}) \approx -(\frac{k\Gamma(\frac{m-1}{2}+1)}{\pi^{(m-1)/2}n})^{\frac{1}{m-1}}(p(\boldsymbol{x}))^{-\frac{1}{m-1}}$$

Recall that the CIL methods based on the TIL+OOD paradigm (i.e., MORE and ROW) use $S_{MD}(\boldsymbol{x})$ to compute the task-prediction probability. As analyzed above, the MD score is in fact IND density estimator, which means $S_{MD}(\boldsymbol{x})$ measures the likelihood of the task distribution $\mathcal{P}_t$. Therefore, the TIL+OOD methods ignores the likelihood of the distribution of other tasks ($\mathcal{P}_{t^c}$), which may fail to make the accurate task prediction. We put the detailed analysis in Appendix H.

## F    ADDITIONAL DETAILS ABOUT TPL

### F.1    COMPUTATION OF THE MD SCORE

Mahalanobis distance score (MD) is an OOD score function initially proposed by Lee et al. (2018b), which is defined as:

$$S_{MD}(\boldsymbol{x}) = 1 / \min_{c \in \mathcal{Y}} \left( (h(\boldsymbol{x}) - \boldsymbol{\mu}_c)^T \boldsymbol{\Sigma}^{-1} (h(\boldsymbol{x}) - \boldsymbol{\mu}_c) \right),$$

where $h(\boldsymbol{x})$ is the feature extractor of a tested OOD detection model $\mathcal{M}$, $\boldsymbol{\mu}_c$ is the centroid for a class $c$ and $\boldsymbol{\Sigma}$ is the covariance matrix. The estimations of $\boldsymbol{\mu}_c$ and $\boldsymbol{\Sigma}$ are defined by

$$\boldsymbol{\mu}_c = \frac{1}{N_c} \sum_{\boldsymbol{x} \in \mathcal{D}_{train}^c} h(\boldsymbol{x}),$$

$$\boldsymbol{\Sigma} = \frac{1}{N} \sum_{c \in |\mathcal{Y}|} \sum_{\boldsymbol{x} \in \mathcal{D}_{train}^c} (h(\boldsymbol{x}) - \boldsymbol{\mu}_c)(h(\boldsymbol{x}) - \boldsymbol{\mu}_c)^T,$$

where $\mathcal{D}_{train}^c := \{\boldsymbol{x} : (\boldsymbol{x}, y) \in \mathcal{D}_{train}, y = c\}$, $\mathcal{D}_{train}$ is the training set, $N$ is the total number of training samples, and $N_c$ is the number of training samples belonging to class $c$.

In the CIL setting, we have to compute $\boldsymbol{\mu}_c$ and $\boldsymbol{\Sigma}$ for each task (assuming all classes in the task have the same covariance matrix) with trained task-specific model $\mathcal{M}^{(t)}$. Specifically, after training on the $t$-th task dataset $\mathcal{D}^{(t)}$, we compute:

$$\boldsymbol{\mu}_c^{(t)} = \frac{1}{N_c} \sum_{(\boldsymbol{x}, c) \in \mathcal{D}^{(t)}} h(\boldsymbol{x}; \phi^{(t)}), \quad \forall c \in \mathcal{Y}^{(t)} \tag{18}$$

$$\boldsymbol{\Sigma}^{(t)} = \frac{1}{|\mathcal{D}^{(t)}|} \sum_{c \in |\mathcal{Y}^{(t)}|} \sum_{(\boldsymbol{x}, c) \in \mathcal{D}^{(t)}} (h(\boldsymbol{x}; \phi^{(t)}) - \boldsymbol{\mu}_c^{(t)})(h(\boldsymbol{x}; \phi^{(t)}) - \boldsymbol{\mu}_c^{(t)})^T \tag{19}$$

We put the memory budget analysis of the saved class centroids $\boldsymbol{\mu}_c^{(t)}$ and co-variance matrices $\boldsymbol{\Sigma}^{(t)}$ in Appendix I.3.

### F.2    COMPUTATION OF LOGIT-BASED SCORES

In Section 5.3, we compare the performance of TPL with different logit-based scores MSP, EBO, and MLS. They are defined as follows:

$$S_{MSP}^{(t)}(\boldsymbol{x}) = \max_{j=1}^{|\mathcal{Y}_t|} softmax \left( f(h(\boldsymbol{x}; \phi^{(t)}); \theta^{(t)}) \right) \tag{20}$$

$$S_{EBO}^{(t)}(\boldsymbol{x}) = \log \sum_{j=1}^{|\mathcal{Y}_t|} \left( \exp \left\{ f(h(\boldsymbol{x}; \phi^{(t)}); \theta^{(t)}) \right\} \right) \tag{21}$$

$$S_{MLS}^{(t)}(\boldsymbol{x}) = \max_{j=1}^{|\mathcal{Y}_t|} \left( f(h(\boldsymbol{x}; \phi^{(t)}); \theta^{(t)}) \right) \tag{22}$$

In traditional OOD detection works, they are tested effective in estimating the probability of "$\boldsymbol{x}$ belongs to IND classes". Thus we can adopt them to design our TPL method to estimate the probability of "$\boldsymbol{x}$ belongs to task $t$".

### F.3    PSEUDO-CODE

To improve reproducibility, we provide the detailed pseudo-code for computing $S_{TPL}(\boldsymbol{x})$ (using Equation (9) in the main text) as Algorithm 1. Then we give the pseudo-code for the CIL training as Algorithm 2 and testing for TPL as Algorithm 3.

---

**Algorithm 1** Compute `TPL` Score with the $t$-th Task-specific Model $\mathcal{M}^{(t)}$

---

**Input:** $Buf^\star$: replay buffer data without classes of task $t$; $\boldsymbol{x}$: test sample; $t$: task-id; $\mathcal{M}^{(t)}$: the trained $t$-th task model $\mathcal{M}^{(t)}$ with feature extractor $h(\boldsymbol{x}; \phi^{(t)})$ and classifier $f(\boldsymbol{x}; \theta^{(t)})$; $\{\boldsymbol{\mu}_c^{(t)}\}_{c \in \mathcal{Y}^{(t)}}$: pre-computed class centroids for task $t$; $\boldsymbol{\Sigma}^{(t)}$: pre-computed covariance matrix for task $t$; $k$: KNN hyper-parameter; $1/\beta_1^{(t)}$: pre-computed empirical mean of $S_{MLS}(\boldsymbol{x})$; $1/\beta_2^{(t)}$: pre-computed empirical mean of $S_{MD}(\boldsymbol{x})$.

**Return:** `TPL` Score $S_{TPL}(\boldsymbol{x})$

1: $S_{MLS}(\boldsymbol{x}) \leftarrow \max_{c \in \mathcal{Y}^{(t)}} f(h(\boldsymbol{x}; \phi^{(t)}); \theta^{(t)})_c$
2: $S_{MD}(\boldsymbol{x}) \leftarrow 1/\left(\min_{c \in \mathcal{Y}^{(t)}}((h(\boldsymbol{x}; \phi^{(t)}) - \boldsymbol{\mu}_c)^T \boldsymbol{\Sigma}^{-1}(h(\boldsymbol{x}; \phi^{(t)}) - \boldsymbol{\mu}_c))\right)$
3: $S_{MLS}(\boldsymbol{x}) \leftarrow S_{MLS}(\boldsymbol{x}) * \beta_1^{(t)}$
4: $S_{MD}(\boldsymbol{x}) \leftarrow S_{MD}(\boldsymbol{x}) * \beta_2^{(t)}$
5: $\boldsymbol{z} \leftarrow h(\boldsymbol{x}; \phi^{(t)})/||h(\boldsymbol{x}; \phi^{(t)})||_2$
6: **for** $\hat{\boldsymbol{x}}_i$ in $Buf^\star$ **do**
7: $\quad \boldsymbol{z}_i \leftarrow h(\hat{\boldsymbol{x}}_i; \phi^{(t)})/||h(\hat{\boldsymbol{x}}_i; \phi^{(t)})||_2$
8: $\quad d_i \leftarrow ||\boldsymbol{z}_i - \boldsymbol{z}||_2$
9: **end for**
10: $\{d_{i_j}\}_{j=1}^{|Buf^\star|} \leftarrow sorted(\{d_i\}_{i=1}^{|Buf^\star|})$
11: $S_{TPL}(\boldsymbol{x}) \leftarrow -\log[\exp\{-S_{MLS}(\boldsymbol{x})\} + \exp\{-S_{MD}(\boldsymbol{x}) - d_{i_k}\}]$

---

**Algorithm 2** CIL Training with `TPL`

---

1: Initialize an empty replay buffer *Buf*
2: **for** training data $\mathcal{D}^{(t)}$ of each task **do**
3: $\quad$ **for** each batch $(\boldsymbol{x}_j, y_j) \subset \mathcal{D}^{(t)} \cup Buf$, until converge **do**
4: $\quad\quad$ Minimize Equation (2) (in the main text) and update the parameters with `HAT`
5: $\quad$ **end for**
6: $\quad$ Compute $\{\boldsymbol{\mu}_c^{(t)}\}_{c \in \mathcal{Y}^{(t)}}$ using Equation (18)
7: $\quad$ Compute $\boldsymbol{\Sigma}^{(t)}$ using Equation (19)
8: $\quad$ Compute $1/\beta_1^{(t)}, 1/\beta^{(t)}$ using Equation (10)
9: $\quad$ Update *Buf* with $\mathcal{D}^{(t)}$
10: **end for**
11: Train the calibration parameters $\{(\tilde{\sigma}_1^{(t)}, \tilde{\sigma}_2^{(t)})\}_{t=1}^T$ following Appendix B

---

**Algorithm 3** CIL Testing with `TPL`

---

**Input**: test sample $\boldsymbol{x}$
**Return:** predicted class $\hat{c}$

1: Compute $S_{TPL}(\boldsymbol{x}; t)$ with each task model $\mathcal{M}^{(t)}$ using Algorithm 1.
2: Preicition with Equation (3): $\hat{c} = \arg\max_{c,t} \sigma_1^{(t)} \cdot \left[softmax\left(f(h(\boldsymbol{x}; \phi^{(t)}); \theta^{(t)})\right)\right]_j \cdot S(\boldsymbol{x}; t) + \sigma_2^{(t)}$

---

# G    DETAILS OF HAT

## G.1    TRAINING

For completeness, we briefly describe the hard attention mechanism of HAT (Serra et al., 2018) used in TPL. In learning the task-specific model $\mathcal{M}^{(t)}$ for each task, TPL at the same time trains a *mask* for each adapter layer. To protect the shared feature extractor from previous tasks, their masks are used to block those important neurons so that the new task learning will not interfere with the parameters learned for previous tasks. The main idea is to use sigmoid to approximate a 0-1 gate function as *hard attention* to mask or unmask the information flow to protect parameters learned for each previous task.

The hard attention at layer $l$ and task $t$ is defined as:

$$\boldsymbol{a}_l^{(t)} = sigmoid(s \cdot \boldsymbol{e}_l^{(t)}),$$

where $s$ is a temperature scaling term, $sigmoid(\cdot)$ denotes the sigmoid function, and $\boldsymbol{e}_l^{(t)}$ is a *learnable* embedding for task $t$. The attention is element-wise multiplied to the ouptut $\boldsymbol{h}_l$ of layer $l$ as

$$\boldsymbol{h}_l' = \boldsymbol{a}_l^{(t)} \otimes \boldsymbol{h}_l$$

The sigmoid function converges to a 0-1 binary gate as $s$ goes to infinity. Since the binary gate is not differentiable, a fairly large $s$ is chosen to achieve a differential pseudo gate function. The pseudo binary value of the attention determines how much information can flow forward and backward between adjacent layers. Denote $\boldsymbol{h}_l = ReLU(\boldsymbol{W}_l \boldsymbol{h}_{l-1} + \boldsymbol{b}_l)$, where $ReLU(\cdot)$ is the rectifier function. For neurons of attention $\boldsymbol{a}_l^{(t)}$ with zero values, we can freely change the corresponding parameters in $\boldsymbol{W}_l$ and $\boldsymbol{b}_l$ without interfering the output $\boldsymbol{h}_l'$. The neurons with non-zero mask values are necessary to perform the task, and thus need a protection for *catastrophic forgetting* (CF).

Specifically, during learning task $t$, we modify the gradients of parameters that are important in performing the previous tasks $1, 2, ..., t-1$ so they are not interfered. Denote the accumulated mask by

$$\boldsymbol{a}_l^{(<t)} = \max(\boldsymbol{a}_l^{(<t-1)}, \boldsymbol{a}_l^{(t-1)}),$$

where $\max(\cdot, \cdot)$ is an element-wise maximum and the initial mask $\boldsymbol{a}_l^{(0)}$ is defined as a zero vector. $\boldsymbol{a}_l^{(<t)}$ is a collection of mask values at layer $l$ where a neuron has value 1 if it has ever been activated previously. The gradient of parameter $w_{ij,l}$ is modified as

$$\nabla w_{ij,l}' = (1 - \min(a_{i,l}^{(<t)}, a_{j,l-1}^{(<t)}))\nabla w_{ij,l},$$

where $a_{i,l}^{(<t)}$ is the $i$-th unit of $\boldsymbol{a}_l^{(<t)}$. The gradient flow is blocked if both neurons $i$ in the current layer and $j$ in the previous layer have been activated. We apply the mask for all layers of adapters except the last layer. The parameters in last layer do not require protection as they are task-specific parameters.

A regularization is introduced to encourage sparsity in $\boldsymbol{a}_l^{(t)}$ and parameter sharing with $\boldsymbol{a}_l^{(<t)}$. The capacity of a network depletes when $\boldsymbol{a}_l^{(<t)}$ becomes an all-one vector in all layers. Despite a set of new neurons can be added in network at any point in training for more capacity, we utilize resources more efficiently by minimizing the loss:

$$\mathcal{L}_{reg} = \frac{\sum_l \sum_i a_{i,l}^{(t)}(1 - a_{i,l}^{(<t)})}{\sum_l \sum_i (1 - a_{i,l}^{(<t)})},$$

The intuition of this term is to regularize the number of masked neurons. Then the loss of HAT defined as the second term from the R.H.S. of Eq (7) in the main text is:

$$\mathcal{L}_{HAT} = \mu \cdot \mathcal{L}_{reg},$$

where $\mu$ is a hyper-parameter to balance the optimization of classification objective and HAT regularization.

### G.2 INFERENCE

In CIL without a task identifier during inference, we are required to forward input data across each task to derive task-specific features $h^{(t)}$ for each task $t = 1, 2, \cdots, T$. This approach can result in computation overhead, especially with extended task sequences. This section proposes to use parallel computing (PC) to mitigate this by achieving comparable time efficiency to one-pass CL methods, albeit with a trade-off of increased memory usage by a factor of $T$ compared to the one-pass CL methods for latent feature storage, a vector of 384 floating point numbers.

Consider the model $\mathcal{M}$ comprising two components: a feature extractor $h(\cdot)$ and a classifier $f(\cdot)$. Unlike the standard model, where both extractor and classifier are universal across tasks, our model uses a shared feature extractor with task-specific classifiers for each task. We analyze the computational costs for each component separately.

For the feature extractor, break it down into $L$ layers. Each layer $l$ involves an affine transformation (with weight $\boldsymbol{W}_l$ and bias $\boldsymbol{b}_l$) followed by an activation function, specifically $ReLU(\cdot)$.[8] In the standard model, computation at each layer $l$ follows:

$$\boldsymbol{h}_l = ReLU(\boldsymbol{W}_l \boldsymbol{h}_{l-1} + \boldsymbol{b}_l) \tag{23}$$

For our model, the computation is extended to:

$$\boldsymbol{h}_l^{(t)} = \boldsymbol{a}_l^{(t)} \otimes ReLU(\boldsymbol{W}_l \boldsymbol{h}_{l-1}^{(t)} + \boldsymbol{b}_l), \quad t = 1, 2, \cdots, T \tag{24}$$

Here, $\boldsymbol{a}_l^{(t)}$ denotes the stored hard attention at layer $l$ for task $t$. Comparing the equations, the additional operation in our method is the element-wise product, which is efficiently parallelizable by vectorizing the sets $\{\boldsymbol{h}_l^{(t)}\}_{t=1}^T$, $\{\boldsymbol{h}_{l-1}^{(t)}\}_{t=1}^T$, and $\{\boldsymbol{a}_l^{(t)}\}_{t=1}^T$ into matrices $\boldsymbol{H}_l$, $\boldsymbol{H}_{l-1}$, and $\boldsymbol{A}_l$. This parallelization achieves near-equivalent time consumption to the standard model, with the trade-off of a $T$-fold increase in memory usage as our feature $\boldsymbol{H}_l$ and $\boldsymbol{H}_{l-1}$ are $T$ times larger compared to $\boldsymbol{h}_l$ and $\boldsymbol{h}_{l-1}$.

For the classifier, each task employs a simple affine transformation (weight $\boldsymbol{W}^{(t)}$ and bias $\boldsymbol{b}^{(t)}$). The standard model uses $\boldsymbol{W} \in \mathbb{R}^{H \times C}$ and $\boldsymbol{b} \in \mathbb{R}^C$, where $H$ and $C$ represent the hidden size and class count, respectively. The standard model's computation is:

$$logits = \boldsymbol{W} \boldsymbol{h}_L + \boldsymbol{b} \tag{25}$$

In contrast, our method involves $T$ task-specific classifiers, each with weight $\boldsymbol{W}^{(t)} \in \mathbb{R}^{H \times \frac{C}{T}}$ and bias $\boldsymbol{b}^{(t)} \in \mathbb{R}^{\frac{C}{T}}$. The task-specific logits are calculated as:

$$logits^{(t)} = \boldsymbol{W}^{(t)} \boldsymbol{h}_L^{(t)} + \boldsymbol{b}^{(t)} \tag{26}$$

By vectorizing task-specific weights, features, and biases, time consumption is maintained at levels comparable to the standard model, and memory consumption remains unaffected.

In conclusion, our approach achieves similar time efficiency as the standard model in CL inference, at the expense of increased runtime memory for storing task-specific features. This trade-off also allows flexibility between memory usage and time consumption by adjusting the **level of parallelism**, offering a balance between runtime duration and memory requirements. For example, one can reduce the running time memory by forwarding the input more times to the model with a lower parallelism level each time.

### G.3 REMARKS

It is important to note that our method can also leverage some other TIL methods to prevent CF other than `HAT`. The reason we chose `HAT` is to make it easy to compare with previous techniques (e.g.,

---

[8] This aligns with our adapter implementation approach.

`MORE` and `ROW`) based on the same setting. For example, we can also exploit `SupSup` (Wortsman et al., 2020), which has already been applied in Kim et al. (2022b) to construct an effective TIL+OOD method and the empirical performance are similar to that using `HAT`. Furthermore, our method is also compatible with other architecture-based TIL methods such as `PNN` (Rusu et al., 2016), `PackNet` (Mallya & Lazebnik, 2018) or Ternary Masks (Masana et al., 2021).

## H    VISUALIZATION OF TASK DISTRIBUTION

The key theoretical analysis behind TPL, theorems 4.1 and 4.2, suggest that an accurate estimate of $\mathcal{P}_{t^c}$ (the distribution of the other tasks, i.e., the complement) is important. If we only estimate $\mathcal{P}_t$ (the feature distribution of task $t$) when doing task prediction and assume $\mathcal{P}_{t^c}$ to be a uniform distribution (which was done by existing methods as analyzed in Appendix E.4), there will be potential risks as shown by the toy example on 1D Gaussian in Section 4.1.

Intuitively, the failure happens when $\mathcal{P}_t$ and $\mathcal{P}_{t^c}$ have some overlap. A test case $\boldsymbol{x}_1$ may have higher likelihood in $\mathcal{P}_t$ than another test case $\boldsymbol{x}_2$, but if it gets even higher likelihood in $\mathcal{P}_{t^c}$, then $\boldsymbol{x}_1$ will be less likely to be drawn from $\mathcal{P}_t$. In this section, we plot $\mathcal{P}_t$ and $\mathcal{P}_{t^c}$ to demonstrate this phenomenon.

Recall that in our CIL scenario, we compute the likelihood ratio for each task to estimate the task prediction probability. For each task $t$, we compute the likelihood of input data $\boldsymbol{x}$ under the task distribution $\mathcal{P}_t$, and use the saved replay data from the other tasks to estimate $\mathcal{P}_{t^c}$. Notice that as we do not conduct task prediction in training, all the discussion here is about inference or test time and we have the small among of saved data from all the learned tasks in the memory buffer. Our goal is to analyze whether $\mathcal{P}_t$ and $\mathcal{P}_{t^c}$ have some overlap.

Specifically, we draw the feature distribution of the five tasks on C10-5T after all tasks are learned under the pre-training setting. To facilitate an accurate estimation of $\mathcal{P}_t$ and $\mathcal{P}_{t^c}$, we use the whole training set of CIFAR-10 to prepare the extracted feature as it contains more data. The vanilla features are high-dimensional vectors in $\mathbb{R}^{384}$, and we use Principled Component Analysis (PCA) (Wold et al., 1987) to project the vectors into $\mathbb{R}^2$. We then use Kernel Density Estimation (KDE) (Terrell & Scott, 1992) to visualize the distribution (density) of the data. The figures are shown as follows:

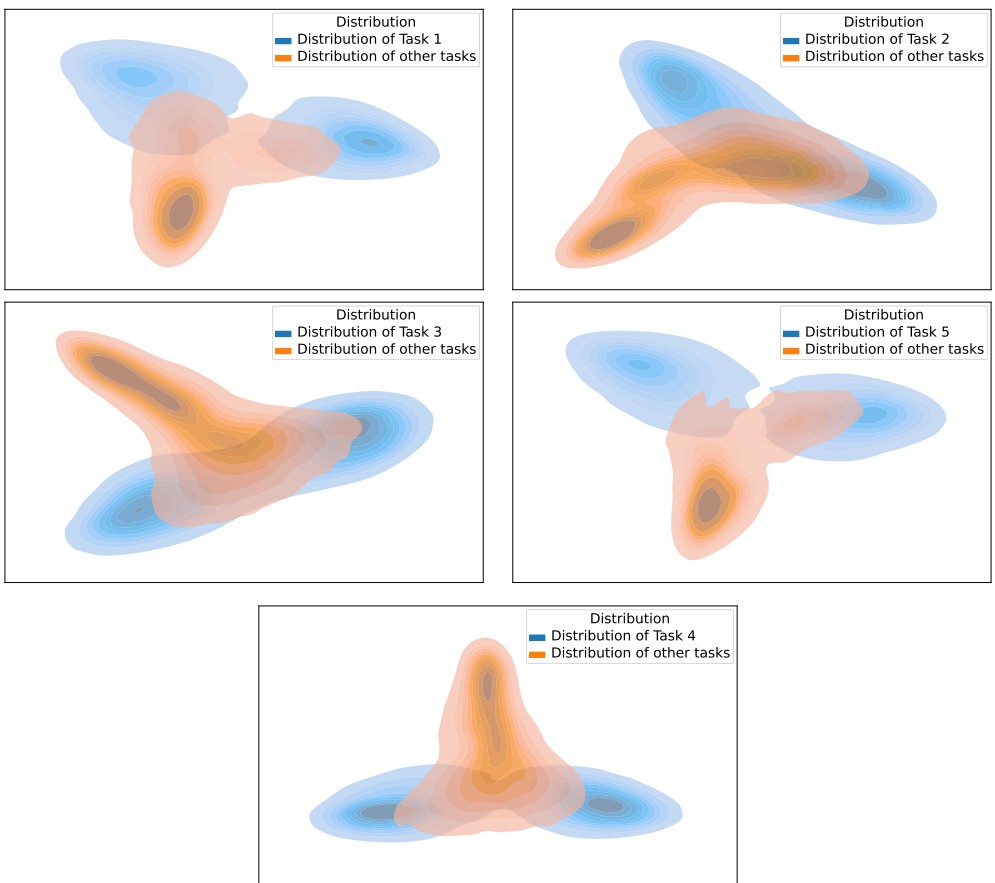

Figure 4: Visualization of feature distribution of Task $t$ ($t$=1,2,3,4,5) data and the other 4 tasks. We use the trained task-specific feature extractor $h(\boldsymbol{x}; \phi^{(t)})$ to extract features from the the training data that belongs to task $t$ (which represent $\mathcal{P}_t$) and the training data that belongs to the other 4 tasks (which represent $\mathcal{P}_{t^c}$).

Notice that the task distributions $\{\mathcal{P}_t\}$ are bimodal as there are two classes in each task in C10-5T. The most interesting observation is that the distribution of the other tasks $\{\mathcal{P}_{t^c}\}$ have overlap with the task distribution $\{\mathcal{P}_t\}$. This indicates that the failure may happen as we analyzed above. For example, we draw a demonstrative failure case of the existing TIL+OOD methods (i.e., MORE and ROW) in Task 1 prediction in Figure 5. In this Figure, TIL+OOD methods will compute the task prediction probability solely based on the high likelihood of $\mathcal{P}_t$ ($t = 1$), while our TPL will consider both high likelihood of $\mathcal{P}_t$ and low likelihood of $\mathcal{P}_{t^c}$. In this case, the red star has higher likelihood in $\mathcal{P}_t$ than the green star (0.9 v.s. 0.4). However, the likelihood ratio between $\mathcal{P}_t$ and $\mathcal{P}_{t^c}$ of the red star is lower than the green star (3 v.s. 20). Therefore, using likelihood ratio between $\mathcal{P}_t$ and $\mathcal{P}_{t^c}$ is crucial in estimating the task prediction probability.

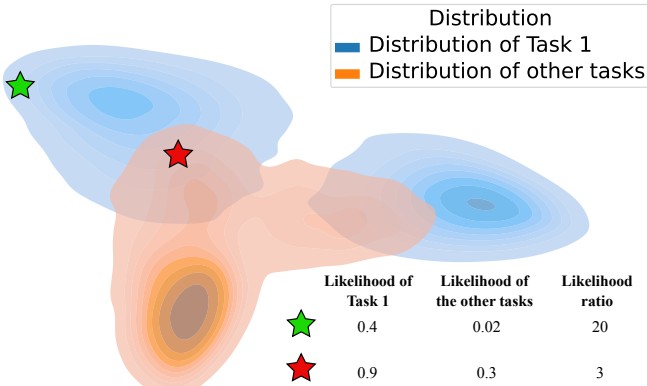

Figure 5: A failure case of TIL+OOD methods that predict the task based on the likelihood of $\mathcal{P}_t$ (e.g., MORE and ROW). In the figure, the red star has higher likelihood in $\mathcal{P}_t$ ($t = 1$) than the green star. However, the likelihood ratio between $\mathcal{P}_t$ and $\mathcal{P}_{t^c}$ of the red star is lower than the green star. The correct choice is to accept the green star to be from Task 1 instead of the red star.

## I    IMPLEMENTATION DETAILS, NETWORK SIZE AND RUNNING TIME

### I.1    IMPLEMENTATION DETAILS OF BASELINES

**Datasets**. We use three popular image datasets. (1) **CIFAR-10** (Krizhevsky & Hinton, 2010) consists of images of 10 classes with 50,000 / 10,000 training / testing samples. (2) **CIFAR-100** (Krizhevsky et al., 2009) consists of images with 50,000 / 10,000 training / testing samples. (3) **Tiny-ImageNet** (Le & Yang, 2015) has 120,000 images of 200 classes with 500 / 50 images per class for training / testing.

**Implementation of CIL baselines (pre-trained setting).** For CIL baselines, we follow the experiment setups as reported in their official papers unless additionally explained in Section 5.1. For the regularization hyper-parameter $\mu$ and temperature annealing term $s$ (see Appendix G) used in HAT, we follow the baseline MORE and use $\mu = 0.75$ and $s = 400$ for all experiments as recommended in (Serra et al., 2018). For the approaches based on network expansion (DER, BEEF, FOSTER), we expand the network of adapters when using a pre-trained backbone. For ADAM, we choose the ADAM(adapter) version in their original paper, which is the best variant of ADAM. As some of the baselines are proposed to continually learn from scratch, we carefully tune **their hyper-parameters** to improve the performance to ensure a fair comparison. The implementation details of baselines under non-pre-training setting are shown in Appendix D.2.

**Hyper-parameter tuning.** Apart from $\beta_1$ and $\beta_2$ discussed in Section 5.3, the only hyper-parameters used in our method TPL are $\gamma$, which is the temperature parameter for task-id prediction, and $k$, which is the hyper-parameter of $d_{KNN}(\boldsymbol{x}, Buf^\star)$ in Equation (7). The value of $\gamma$ and $k$ are searched from $\{0.01, 0.05, 0.10, 0.50, 1.0, 2.0, 5.0, 10.0\}$ and $\{1, 2, 5, 10, 50, 100\}$, respectively. We choose $\gamma = 0.05$, and $k = 5$ for all the experiments as they achieve the overall best results.

**Training Details.** To compare with the strongest baseline MORE and ROW, we follow their setup (Kim et al., 2022a; 2023) to set the training epochs as 20, 40, 15, 10 for CIFAR-10, CIFAR-100, T-5T, T-10T respectively. And we follow them to use SGD optimizer, the momentum of 0.9, the batch size of 64, the learning rate of 0.005 for C10-5T, T-5T, T-10T, C100-20T, and 0.001 for C100-10T.

### I.2    HARDWARE AND SOFTWARE

We run all the experiments on NVIDIA GeForce RTX-2080Ti GPU. Our implementations are based on Ubuntu Linux 16.04 with Python 3.6.

### I.3    COMPUTATIONAL BUDGET ANALYSIS

#### I.3.1    MEMORY CONSUMPTION

We present the network sizes of the CIL systems (with a pre-trained network) after learning the final task in Table 12.

With the exception of SLDA and L2P, all the CIL methods we studied utilize trainable adapter modules. The transformer backbone consumes 21.6 million parameters, while the adapters require 1.2M for CIFAR-10 and 2.4M for other datasets. In the case of SLDA, only the classifier on top of the fixed pre-trained feature extractor is fine-tuned as it requires a fixed feature extractor for all tasks, while L2P maintains a prompt pool with 32k parameters. Additionally, each method requires some specific elements (e.g., task embedding for HAT), resulting in varying parameter requirements for each method.

As mentioned in Appendix F.1, our method also necessitates the storage of class centroids and covariance matrices. For each class, we save a centroid of dimension 384, resulting in a total of 3.84k, 38.4k, and 76.8k parameters for CIFAR-10, CIFAR-100, and TinyImageNet, respectively. The covariance matrix is saved per task, with a size of $384 \times 384$. Then the parameter count of covariance matrix can be computed by $T \times 384 \times 384$ for a dataset with $T$ tasks. Consequently, the total parameter count is 737.3k, 1.5M, 2.9M, 737.3k, and 1.5M for C10-5T, C100-10T, C100-20T, T-5T, and T-10T, respectively. It is worth noting that this consumption is relatively small when compared to certain replay-based methods like iCaRL and HAL, which require a teacher model of the same size as the training model for knowledge distillation.

Table 12: Network size measured in the number of parameters (# parameters) for each method without the memory buffer.

|  | C10-5T | C100-10T | C100-20T | T-5T | T-10T |
|---|---|---|---|---|---|
| OWM | 24.1M | 24.4M | 24.7M | 24.3M | 24.4M |
| ADAM | 22.9M | 24.1M | 24.1M | 24.1M | 24.1M |
| PASS | 22.9M | 24.2M | 24.2M | 24.3M | 24.4M |
| $HAT_{CIL}$ | 24.1M | 24.4M | 24.7M | 24.3M | 24.4M |
| SLDA | 21.6M | 21.6M | 21.6M | 21.7M | 21.7M |
| L2P | 21.7M | 21.7M | 21.7M | 21.8M | 21.8M |
| iCaRL | 22.9M | 24.1M | 24.1M | 24.1M | 24.1M |
| A-GEM | 26.5M | 31.4M | 31.4M | 31.5M | 31.5M |
| EEIL | 22.9M | 24.1M | 24.1M | 24.1M | 24.1M |
| GD | 22.9M | 24.1M | 24.1M | 24.1M | 24.1M |
| DER++ | 22.9M | 24.1M | 24.1M | 24.1M | 24.1M |
| HAL | 22.9M | 24.1M | 24.1M | 24.1M | 24.1M |
| DER | 27.7M | 45.4M | 69.1M | 33.6M | 45.5M |
| FOSTER | 28.9M | 46.7M | 74.2M | 35.8M | 48.1M |
| BEEF | 30.4M | 48.4M | 82.3M | 37.7M | 50.6M |
| MORE | 23.7M | 25.9M | 27.7M | 25.1M | 25.9M |
| ROW | 23.7M | 26.0M | 27.8M | 25.2M | 26.0M |
| **TPL** | 23.7M | 25.9M | 27.7M | 25.1M | 25.9M |

### I.3.2 RUNNING TIME

The computation for our method TPL is very efficient. It involves standard classifier training and likelihood raio score computation, which employed some OOD detection methods. The OOD score computation only involves mean, covariance computation, and KNN search, which are all very efficient with the Python packages scikit-learn[9] and faiss[10]. We give the comparison in running time in Table 13. We use HAT as the base as MORE, ROW and TPL all make use of HAT and MORE and ROW are the strongest baselines.

Table 13: Average running time measured in minutes per task (min/T) for three systems.

|  | C10-5T | C100-10T | C100-20T | T-5T | T-10T |
|---|---|---|---|---|---|
| $HAT_{CIL}$ | 17.8 min/T | 17.6 min/T | 9.4 min/T | 28.0 min/T | 9.48 min/T |
| MORE | 20.6 min/T | 23.3 min/T | 11.7 min/T | 32.8 min/T | 11.2 min/T |
| ROW | 21.8 min/T | 25.2 min/T | 12.6 min/T | 34.1 min/T | 11.9 min/T |
| TPL | 20.7 min/T | 23.3 min/T | 11.7 min/T | 32.5 min/T | 11.2 min/T |

---

[9] https://scikit-learn.org/stable/
[10] https://github.com/facebookresearch/faiss

## J    LIMITATIONS

Here we discuss the limitations of our proposed method `TPL`.

**First,** our `TPL` method relies on the saved data in the memory buffer like traditional replay-based methods, which may have privacy concerns in some situations and also need extra storage. We will explore how to improve task-id prediction in the CIL setting without saving any previous data in our future work. **Second,** `TPL` uses a naive saving strategy that samples task data randomly to put in the memory buffer for simplicity. In our future work, we would also like to consider better buffer saving strategies (Jeeveswaran et al., 2023) and learning algorithm for buffer data (Bhat et al., 2022), which may enable more accurate likelihood ratio computation. **Third,** this paper focuses on the traditional *offline* CIL problem set-up. In this mode, all the training data for each task is available upfront when the task arrives and the training can take any number of epochs. Also, the label space $\mathcal{Y}^{(t)}$ of different tasks are disjoint. In online CIL (Guo et al., 2022), the data comes in a stream and there may not be clear boundary between tasks (called *Blurry Task Setting* (Bang et al., 2022)), where the incoming data labels may overlap across tasks. It is an interesting direction for our future work to explore how to adapt our method to exploit the specific information in this setting.

