# OpenReview forum: "Class Incremental Learning via Likelihood Ratio Based Task Prediction"
_ICLR.cc/2024/Conference — ICLR 2024 poster_

### Official Review · Reviewer_aMSi · 2023-10-24

**Soundness:** 3 good
**Presentation:** 1 poor
**Contribution:** 2 fair
**Rating:** 5
**Confidence:** 4

**Summary:**

In this paper, the author proposes the use of a Likelihood Ratio to identify the Task-id for Class Incremental Learning (CIL). Traditionally, out-of-distribution (OOD) detectors were used for task identification, but this paper introduces a new method called TPLR (Task-id Prediction based on Likelihood Ratio) that leverages additional information like replay data and learned tasks for more effective and principled task-id prediction. TPLR outperforms traditional CIL approaches.

**Strengths:**

The motivation and method sound solid to me. I agree with the author that task-id prediction is pivotal for CIL under specific circumstances.

- Motivation is clear and straightforward:  The author argues that using a traditional OOD detector is not optimal for task predictions and here they leverage the information in the CIL process for task-id prediction.

- The proof and methods in section 3 and 4 look good to me.

**Weaknesses:**

1. The writing requires improvement. The author frequently used abbreviations and jargon, especially in the introduction, which occasionally left me puzzled. It would be beneficial if these terms were interpreted more straightforwardly.

2. The related works are also unclear:
- Although the author clarifies their focus on Class incremental learning, which doesn't provide the task-id during inference, it remains ambiguous whether they are using a memory buffer (rehearsal-based) or are memory-free (online CIL). I suggest the author address this in the introduction and related works.
- Some recent benchmarks are missing: The author left memory-free (non-replay-based approaches) CIL in related works. The author also left balanced CIL works, e.g., SS-IL, TKIL.

3. Experimental settings:
- Table 1 is impressive, but the comparisons seem biased. The author claims they compared with 17 baselines, including 11 replay-based and 6 non-replay-based. From my understanding, the author requires a memory buffer, as indicated in the "Overview of the Proposed TPLR", equation 2.
-  It would be more equitable if the author juxtaposed their method with replay-based CIL. Specifically, the author should draw a clear comparison with methods using task-id prediction, highlighting the advantages of their technique.
- One import baseline is missing: AFC[3]


4. The inference setting remains unclear. Does the author predict both the task-id and class-id simultaneously? Is there any fine-tuning step involved? Typically, some fine-tuning follows the task-id prediction. e.g., iTAML. If the author's method circumvents this, it could be seen as a distinct advantage. Therefore, I recommend the author incorporate a discussion about the computational load when integrating likelihood ratio predictions, elucidating the benefits and drawbacks of this model.

5. Lacks Visualizations: Could the author add a real visualization of data distribution, like the "Feature-based likelihood ratio score" in Figure 1. It will be strong evidence the TPLR works well.


[1] Ss-il: Separated softmax for incremental learning. In Proceedings of the IEEE/CVF International conference on computer vision

[2] TKIL: Tangent Kernel Optimization for Class Balanced Incremental Learning. In Proceedings of the IEEE/CVF International Conference on Computer Vision

[3] Class-incremental learning by knowledge distillation with adaptive feature consolidation. In Proceedings of the IEEE/CVF conference on computer vision and pattern recognition

**Questions:**

Please refer the weakness;

---

> ### Author Response · Authors · 2023-11-16
>
> Dear Reviewer aMSi,
>
> we are glad to address your concerns as follows.
>
> > The writing requires improvement. The author frequently used abbreviations and jargon, especially in the introduction, which occasionally left me puzzled. It would be beneficial if these terms were interpreted more straightforwardly.
>
> Thanks for your suggestion. We have significantly reduced the use of abbreviations and jargon to make the introduction section easier to read and to interpret. Since we edited in many places, we did not highlight the changes. You may check out the new introduction section in the revised paper.
>
> > Although the author clarifies their focus on Class incremental learning, which doesn't provide the task-id during inference, it remains ambiguous whether they are using a memory buffer (rehearsal-based) or are memory-free (online CIL). I suggest the author address this in the introduction and related works.
>
> Yes, we are using a memory buffer to save some previous data as in replay-based methods but the saved data in our method are used to estimate the distribution of U_{CIL} in training each task and also in testing rather than to replay them in training a new task like replay-based methods. We are not doing online CIL. We have added footnote 3 to make these clearer.
>
> > Some recent benchmarks are missing: The author left memory-free (non-replay-based approaches) CIL in related works. The author also left balanced CIL works, e.g., SS-IL, TKIL.
>
> Many systems in the related work section are memory-free methods, e.g., those regularization-based methods, orthogonal-projection based methods, and some of those task-id prediction based methods.
>
> As a large number of papers have been published in continual learning, there are inevitable casualties in literature review. Thanks for pointing out the balanced CIL work. We have added it and cited both SS-IL and TKIL papers in the related work section. Please see the last sentence (in blue) on page 2 of the revised paper.
>
> > Table 1 is impressive, but the comparisons seem biased. The author claims they compared with 17 baselines, including 11 replay-based and 6 non-replay-based. From my understanding, the author requires a memory buffer, as indicated in the "Overview of the Proposed TPLR", equation 2.
>
> We also think that our results are very strong. Thanks. We have some difficulty understanding your comment “the comparisons seem biased”. Could you explain more? Yes, we use a memory buffer to save some past data like replay based methods but we have compared with 11 strong replay-based baselines. We also include 6 non-replay based methods as some non-replay based methods are also strong, e.g., PASS and SLDA.
>
> > It would be more equitable if the author juxtaposed their method with replay-based CIL. Specifically, the author should draw a clear comparison with methods using task-id prediction, highlighting the advantages of their technique.
>
> Yes, we agree with you to juxtapose our method with replay-based CIL, and we have included 11 strong replay-based baselines in Table 1. Besides, the best baselines MORE and ROW also use task-id prediction. The results show that our method TPLR outperforms them by a large margin. We did not include some earlier task-id prediction based CIL methods (but they have been reviewed in the related work section) as they are much weaker than MORE and ROW. Below, we report the results of two more task-id prediction based CIL methods, iTAML [1] and PR-Ent [2], based on the same pre-trained model. Notice that the vanilla iTAML assumes each test batch of many instances comes from a single task and uses the whole batch of data to predict the task-id. We set its test batch size as 1 to disable the assumption for a fair comparison as all other methods predict one test instance at a time, which is more realistic.
>
> |        | C10-5T    | C100-10T  | C100-20T  | T-5T      | T-10T     | Average   |
> | ------ | --------- | --------- | --------- | --------- | --------- | --------- |
> | iTAML  | 78.65     | 59.56     | 52.31     | 49.53     | 46.34     | 57.28     |
> | PR-Ent | 84.37     | 64.51     | 60.52     | 58.63     | 55.44     | 64.69     |
> | TPLR   | **92.33** | **76.53** | **76.34** | **68.64** | **67.20** | **76.21** |
>
> > One import baseline is missing: AFC[3]
>
> We’ve added AFC as a new baseline in Table 1 and cited the paper as well. A summary of the results (accuracy after learning the final task) is presented in the following table. It can be seen that our TPLR also outperforms AFC by a large margin..
>
> |      | C10-5T    | C100-10T  | C100-20T  | T-5T      | T-10T     | Average   |
> | ---- | --------- | --------- | --------- | --------- | --------- | --------- |
> | AFC  | 84.59     | 69.85     | 69.44     | 56.40     | 55.35     | 67.13     |
> | TPLR | **92.33** | **76.53** | **76.34** | **68.64** | **67.20** | **76.21** |

---

> > ### Author Response · Authors · 2023-11-16
> >
> > > The inference setting remains unclear. Does the author predict both the task-id and class-id simultaneously? Is there any fine-tuning step involved? Typically, some fine-tuning follows the task-id prediction. e.g., iTAML. If the author's method circumvents this, it could be seen as a distinct advantage. Therefore, I recommend the author incorporate a discussion about the computational load when integrating likelihood ratio predictions, elucidating the benefits and drawbacks of this model.
> >
> > Yes, we predict both the task-id and class-id simultaneously, more specifically estimating their probabilities. There is no fine-tuning step involved. As shown on the right part of Figure 1, our inference pipeline allows the simultaneous computation of within-task class prediction (top part) and task-id prediction (bottom part) probabilities. We have discussed the computation costs of our method in Appendix G.3, which shows that the likelihood ratio prediction is very efficient. Overall, the time efficiency is comparable with the best accuracy performing baselines MORE and ROW.
> >
> > > Lacks Visualizations: Could the author add a real visualization of data distribution, like the "Feature-based likelihood ratio score" in Figure 1. It will be strong evidence the TPLR works well.
> >
> > That’s a good suggestion! We have added the visualization of data distribution in Appendix F. The visualization demonstrates that TPLR’s likelihood ratio is crucial in estimating the task prediction probability. We put the detailed analysis in Appendix F.
> >
> > [1] Rajasegaran et al. iTAML: An Incremental Task-Agnostic Meta-learning Approach. CVPR 2020.
> >
> > [2] Henning et al. Posterior Meta-Replay for Continual Learning. NeurIPS 2021.
> >
> > Hope our additional experimental results address your concerns. We also hope that our strong results and novel and principled technique can change your mind to support our paper. If you have any further comments, we will be very happy to address them.

---

> > > ### Comment · Reviewer_aMSi · 2023-11-16
> > > **I thank the authors for updating the manuscript with new results and visualizations to clarify my questions.**
> > >
> > > I thank the authors for updating the manuscript with new results and visualizations to clarify my questions. However, the following concerns still exist.
> > >
> > > 1. 'Comparisons seem biased': The authors should organize their writing to reflect a clear and straightforward comparison with proper baseline methods. I suggest the authors first clarify the categories of CL/task-IL/Class-IL and corresponding subcategories (memory/non-memory). From our discussion and the authors' rebuttals, it seems that the proposed method is memory-based class incremental learning, i.e., it needs memory and doesn't need a task label during reference. If that is the case, the authors should only compare it with such settings in CIL. Therefore, adding the memory-free approach to compare with the proposed method is unclear and also unfair. I have also read the other reviewer's comments and I see other reviewer also has a similar question:
> > >
> > > 'In Table 1, which of these results are using the task label at inference time? For example, HAT needs the task label. So, are the results of HAT comparable here with the other methods? Or is HAT having a forward pass with each task label and then using some heuristic to pick the class?'"
> > >
> > > However, the author's reply also makes me more confused:
> > >
> > > "In Table 1, our method using HAT needs the task label. But since we run all task models at test time to compute the task label probability for each task, effectively we do not need any task label."
> > >
> > > Therefore, it is unclear to me whether the proposed method requires task labels or not.
> > >
> > > 2. I also checked Appendix G as the author suggested. Table 8 is unclear to me. If the author is seeking a fair comparison, they are supposed to use the same model for comparisons. Why are there differences in parameters for different methods?

---

> ### Author Response · Authors · 2023-11-17
>
> Dear Reviewer aMSi,
>
> Thank you very much for your feedback. We are sorry for the confusion. Please see our clarifications below.
>
> > Comment: I thank the authors for updating the manuscript with new results and visualizations to clarify my questions. However, the following concerns still exist.
> >
> > 'Comparisons seem biased': The authors should organize their writing to reflect a clear and straightforward comparison with proper baseline methods. I suggest the authors first clarify the categories of CL/task-IL/Class-IL and corresponding subcategories (memory/non-memory). From our discussion and the authors' rebuttals, it seems that the proposed method is memory-based class incremental learning, i.e., it needs memory and doesn't need a task label during reference. If that is the case, the authors should only compare it with such settings in CIL. Therefore, adding the memory-free approach to compare with the proposed method is unclear and also unfair.
>
> **Answer**: We are happy to remove memory-free methods from the table.
>
> > I have also read the other reviewer's comments and I see other reviewer also has a similar question: 'In Table 1, which of these results are using the task label at inference time?”
>
> **Answer**: None of the methods in Table 1 uses the task label at inference time.
>
> > For example, HAT needs the task label. So, are the results of HAT comparable here with the other methods? Or is HAT having a forward pass with each task label and then using some heuristic to pick the class?
>
> > However, the author's reply also makes me more confused:
>
> > "In Table 1, our method using HAT needs the task label. But since we run all task models at test time to compute the task label probability for each task, effectively we do not need any task label."
>
> **Answer**: Sorry for the confusion. Let us try to clarify this.
>
> The original HAT needs the task label for each test instance for task-incremental learning (TIL). However, we are using HAT for class-incremental learning (CIL) and we have **no** task label for each test instance x, so we need to adapt HAT for CIL (as we explained in the footnote of page 7).
>
> Specifically, we run x on the model of every task. For example, we have 4 tasks and we have learned 4 models for the 4 tasks based on HAT: Model1, Model2, Model3, Model4. HAT’s results in Table 1 are obtained by running x on every model via a forward pass to compute the following (for example):
>
> For Model1: class c5 in task 1 gives the highest softmax value of 0.7
>
> For Model2: class c2 in task 2 gives the highest softmax value of 0.6
>
> For Model3: class c6 in task 3 gives the highest softmax value of 0.3
>
> For Model4: class c1 in task 4 gives the highest softmax value of 0.9
>
> Then the test instance x is assigned the final class of c1 in task 4 because its highest softmax value is the highest among the 4 task models.
>
> For our method, we do similarly but we compute the task probability based on each model with a softmax-based normalization, i.e., P(task-1 | x), P(task-2 | x), P(task-3 | x), P(task-4 | x).
>
> > Therefore, it is unclear to me whether the proposed method requires task labels or not
>
> **Answer**: No, we do not need task labels.
>
> > I also checked Appendix G as the author suggested. Table 8 is unclear to me. If the author is seeking a fair comparison, they are supposed to use the same model for comparisons. Why are there differences in parameters for different methods?
>
> **Answer**: We cannot ensure each method has the same number of network parameters as they are designed with different components in their algorithm. For example, DER, FOSTER, and BEEF are network-expansion-based methods. They dynamically increase the model size during continual learning as more tasks are learned; L2P needs to maintain a prompt pool; OWM is based on orthogonal projection and it needs to save some matrices.
>
> To seek a fair comparison, we can only ensure that they are using the same backbone, i.e., pre-trained DeiT or non-pre-trained ResNet18 in our two experimental settings.
>
> Hope these clear all your doubts. If you have additional questions or comments, please let us know.

---

### Official Review · Reviewer_h9ds · 2023-10-31

**Soundness:** 4 excellent
**Presentation:** 3 good
**Contribution:** 3 good
**Rating:** 8
**Confidence:** 3

**Summary:**

In this paper, the authors address the challenge of task identification (task-id prediction) in Class Incremental Learning (CIL). They propose a novel method named TPLR (Task-id Prediction based on Likelihood Ratio), which enhances task-id prediction by utilizing replay data to estimate the distribution of non-target tasks. This approach allows for a more principled solution compared to traditional Out-of-Distribution (OOD) detection methods that cannot estimate the vast universe of non-target classes due to lack of data.

TPLR calculates the likelihood ratio between the data distribution of the current task and that of its complement, providing a robust mechanism for task-id prediction. The method is integrated into the Hard Attention to the Task (HAT) structure, which employs learned masks to prevent catastrophic forgetting, adapting the architecture to facilitate both task-id prediction and within-task classification.

The authors demonstrate through extensive experimentation that TPLR substantially outperforms existing baselines in CIL settings. This performance is consistent across different configurations, including scenarios with and without pre-trained feature extractors. The paper's contributions offer significant advancements for task-id prediction in CIL, proposing a method that leverages available data more effectively than prior approaches.

**Strengths:**

Originality:

- TPLR's innovation lies in its unique application of likelihood ratios for task-id prediction, an approach that distinctively diverges from traditional OOD detection methods.
- The paper creatively leverages replay data to estimate the data distribution for non-target tasks, which is a novel use of available information in the CIL framework.
- Integration of TPLR with the HAT method showcases an inventive combination of techniques to overcome catastrophic forgetting while facilitating task-id prediction.

Quality:

-The methodological execution of TPLR is of high quality. It is underpinned by a strong theoretical framework that is well-articulated and logically sound.
- Extensive experiments validate the robustness and reliability of TPLR, demonstrating its superiority over state-of-the-art baselines.

Clarity:

The paper writing quality is satisfactory.

Significance:

TPLR's ability to outperform existing baselines marks a significant advancement in the domain of CIL, potentially influencing future research directions and applications.
The paper's approach to using replay data for improving task-id prediction could have broader implications for continual learning paradigms beyond CIL.

**Weaknesses:**

The key weakness of this work I would argue is its overly complex presentation. I find that the organization of the paper can easily distract and confuse the reader, often finding myself fishing for key details of the main method.

**Questions:**

- While the writing quality is satisfactory, I would argue for a friendlier approach to outlining the proposed method. First, outline the key ingredients. Then explain how they interact. Finally cross-reference these with the existing figure.
- The existing figure is a bit too 'noisy' in terms of the information it is showing and the order it is showing it in. Consider reorganizing it so it can be read from left to right, top to bottom and with more emphasis on the key ideas and less detail that can distract from that.

---

> ### Author Response · Authors · 2023-11-16
>
> Dear Reviewer h9ds,
>
> Thank you very much for supporting our paper. Below, we address your comments.
>
> > While the writing quality is satisfactory, I would argue for a friendlier approach to outlining the proposed method. First, outline the key ingredients. Then explain how they interact. Finally cross-reference these with the existing figure.
>
> Thank you for the suggestion. We tried to follow a top-down approach to present our method:
>
> Section 3 presents the motivation and overview of our TPLR method, which consists of training and testing.
>
> - For training, the key components are our proposed network architecture and HAT for protecting each task model, which corresponds to the left part of Figure 1.
> - For testing, there are two key ingredients: within-task prediction and task-id prediction probabilities, which correspond to the top right part and bottom right part of Figure 1.
>
> As estimating task-id prediction probability is the most critical part, we use the whole of Section 4 to introduce the theorem and the design. Also, the design consists of three key ingredients:
>
> - (1) estimating the likelihood ratio between $P_t$ and $P_{t^c}$ (corresponds to the “Feature-based likelihood ratio score” in Figure 1).
> - (2) combining with a logit-based score (corresponds to “Max logit score” in Figure 1).
> - (3) converting task-id prediction scores to probabilities (corresponds to “softmax” in Figure 1).
>
> Following your suggestions, we have revised the paper (especially Section 3 and Section 4) to improve the clarity and also make it friendlier by adding more cross-references (in blue fonts).
>
> > The existing figure is a bit too 'noisy' in terms of the information it is showing and the order it is showing it in. Consider reorganizing it so it can be read from left to right, top to bottom and with more emphasis on the key ideas and less detail that can distract from that.
>
> Thank you for this suggestion. We revised Figure 1 to make it friendlier. We added the subtitles “Network Architecture” and “Inference Pipeline” to divide the figure into two parts (left and right). We will keep thinking about how to make Figure 1 even easier to understand.

---

### Official Review · Reviewer_SU84 · 2023-10-31

**Soundness:** 3 good
**Presentation:** 3 good
**Contribution:** 3 good
**Rating:** 5
**Confidence:** 4

**Summary:**

This paper proposes a novel method for class incremental learning (CIL) by directly predicting the task identifiers to perform the task-wise prediction. Using the energy based models, the given model computes the scores for each task based on the Mahalanobis distance and KNN distance, and estimate the task label. Furthermore, the proposed model actively utilizes the pre-trained model by just training the adapter module to efficiently train the parameters. In the experiment, the given algorithm outperforms the baselines in both CIFAR and Tiny-ImageNet dataset. In addition, the authors show the effectiveness of using each component in the ablation study.

**Strengths:**

1. By directly estimating the task identifier, the proposed algorithm outperforms other baselines in the benchmark dataset.

2. Since the proposed model utilize the task-wise classifier, it can be robust to the class imbalance problem which can occur when the difference between the size  of replay buffer and training data are large.

**Weaknesses:**

1. I wonder the proposed methods can achieve high task-prediction accuracy. Different from the ideal situation, the accuracy may be lower than we expected. if the semantics across different classes are similar, the task-prediction accuracy can be low, and the overall performance also can decrease.

2. Can this method outperform other baselines when it does not use the pre-trained model in ImageNet-1K? Furthermore, if the dataset used for pre-training are randomly selected (i.e. Randomly extract 500 classes from ImageNet-1K), can this method outperform other baselines? Since ImageNet-1K or other large datasets contain similar classes, the task-prediction is much harder than CIFAR or Tiny-ImageNet

**Questions:**

Already mentioned in the Weakness section

---

> ### Author Response · Authors · 2023-11-16
>
> Dear Reviewer SU84,
>
> we are glad to address your concerns as follows.
>
> > I wonder the proposed methods can achieve high task-prediction accuracy. Different from the ideal situation, the accuracy may be lower than we expected. if the semantics across different classes are similar, the task-prediction accuracy can be low, and the overall performance also can decrease.
>
> We are not sure about the first part of your question and “the accuracy we expected.” Our method does not explicitly predict the task, but only estimates the task prediction probability for each test instance. Just for curiosity, we use the estimated task prediction probability to do task prediction by choosing the task with the highest probability. The task prediction accuracy results (after learning the final task) are shown in the table below (we also compare with the best-performing baselines MORE and ROW). Our TPLR consistently outperforms the two strong baselines on task prediction accuracy, which demonstrates the superiority of TPLR over OOD detection-based methods for task prediction in MORE and ROW.
>
> |      | C10-5T | C100-10T | C100-20T | T-5T   | T-10T  | Average |
> | ---- | ------ | -------- | -------- | ------ | ------ | ------- |
> | MORE | 0.8861 | 0.7076   | 0.6997   | 0.7368 | 0.6835 | 0.7427  |
> | ROW  | 0.9131 | 0.7531   | 0.7411   | 0.7382 | 0.6855 | 0.7662  |
> | TPLR | **0.9291** | **0.7819**   | **0.7681**   | **0.7891** | **0.7393** | **0.8015**  |
>
> About the second part of your comment, if the semantics across different classes are similar, the accuracy should be lower, which is the case for any supervised learning. In our case, the accuracy will be lower for all CIL baselines.
>
> > Can this method outperform other baselines when it does not use the pre-trained model in ImageNet-1K? Furthermore, if the dataset used for pre-training are randomly selected (i.e. Randomly extract 500 classes from ImageNet-1K), can this method outperform other baselines? Since ImageNet-1K or other large datasets contain similar classes, the task prediction is much harder than CIFAR or Tiny-ImageNet.
>
> We have conducted the experiment on ImageNet-1K without a pre-trained model. Following the setting in BEEF [1], we split the ImageNet-1K into 10 tasks and compared our TPLR with the top-five baselines DER, FOSTER, BEEF, MORE, and ROW. The CIL accuracy results (after learning the final task) are shown in the table below. We can see that TPLR also outperforms the baselines on ImageNet-1K without a pre-trained model.
>
> | DER    | FOSTER | BEEF   | MORE   | ROW    | TPLR       |
> | ------ | ------ | ------ | ------ | ------ | ---------- |
> | 0.5883 | 0.5853 | 0.5867 | 0.5030 | 0.5214 | **0.6171** |
>
> To address the second part of your comment, we would first like to stress that our pre-trained model did not use the full ImageNet data. Instead, our pre-training used only 611 classes of ImageNet after removing 389 classes that are similar or identical to the classes of the experiment data CIFAR and TinyImageNet to prevent information leak (see the last sentence in page 7).
>
> Based on your suggestion, we also conducted experiments by randomly selecting 500 classes from the remaining 611 classes after removing overlapping classes with CIFAR and TinyImageNet to prevent information leaks. We feel this is the right way to do it. We trained three different pre-trained models using three different sets of 500 classes. The CIL accuracy results (after learning the final task) for the top-five baselines and our TPLR are shown as follows. For simplicity, we average the results on the five datasets (C10-5T, C100-10T, C100-20T, T-5T, T-10T). We observed that TPLR has similar performance gains compared to those reported in the paper.
>
> |                     | DER    | FOSTER | BEEF   | MORE   | ROW    | TPLR       |
> | ------------------- | ------ | ------ | ------ | ------ | ------ | ---------- |
> | pre-trained model 1 | 0.6815 | 0.6817 | 0.7011 | 0.7132 | 0.7312 | **0.7615** |
> | pre-trained model 2 | 0.6801 | 0.6777 | 0.6935 | 0.7081 | 0.7253 | **0.7591** |
> | pre-trained model 3 | 0.6753 | 0.6823 | 0.7015 | 0.7077 | 0.7214 | **0.7585** |
>
> [1] Wang et al. Beef: Bi-compatible class-incremental learning via energy-based expansion and fusion. ICLR 2022.
>
> Hope our additional experimental results address your concerns. We also hope that our strong results and novel and principled technique can change your mind to support our paper. If you have any further comments, we will be very happy to address them.

---

> ### Author Response · Authors · 2023-11-22
> **A Gentle Reminder**
>
> Dear Reviewer SU84,
>
> The deadline of the discussion period is soon approaching. We wonder whether our answers to your questions and the new experimental results based on your suggestion have addressed your concerns. If there are any additional discussion points or questions, we are happy to discuss. Thank you.
>
> Authors

---

> ### Author Response · Authors · 2023-11-23
>
> Dear Reviewer SU84,
>
> Regarding your comment that similar classes across tasks may decrease the accuracy, we have conducted a new experiment, described below.
>
> In our paper, we used the CIFAR-100 dataset, which comprises 100 classes grouped into 20 superclasses, each containing 5 similar classes (e.g., the "flowers" superclass includes "orchids", "poppies", "roses", "sunflowers", "tulips"). We conducted the CIFAR100-20T (20 tasks) experiment using different task sequences:
>
> - seq0: each task contains classes from a distinct superclass, which means that within each task the classes are similar, but across tasks, the classes are dissimilar.
>
> - seq1: each task contains classes from different superclasses, which means that within each task, the classes are dissimilar, but across tasks, there are many similar classes.
>
> - seq2: seq3 and seq4 are randomly-mixed tasks, meaning that the classes of each task are randomly selected from random superclasses.
>
> This design shows different combinations of similar or dissimilar classes in the same or different tasks. As we can see from the table below (the results are the averages of 5 random runs with 5 random seeds), the performances across these task sequences are very similar. The differences are not significant.
>
> | **Last. Acc**               | **seq0** | **seq1** | **seq2** | **seq3** | **seq4** |
> | --------------------------- | -------- | -------- | -------- | -------- | -------- |
> | mean (5 seeds)               | 76.40    | 76.25    | 76.29    | 76.32    | 76.42    |
> | standard deviation (5 seeds) | 0.28     | 0.31     | 0.38     | 0.24     | 0.33     |

---

### Official Review · Reviewer_fLUE · 2023-11-03

**Soundness:** 3 good
**Presentation:** 3 good
**Contribution:** 2 fair
**Rating:** 6
**Confidence:** 4

**Summary:**

The authors propose to use out-of-distribution ideas to solve the gap between task incremental and class incremental learning, indirectly predicting the task label and using it to refine the class prediction. They pose that using a low forgetting method such as HAT and pairing it with a good task-prediction from the ood-inspired setting, allows better estimates of both the intra- and inter-task probabilities, which leads to better performance in CIL scenarios.

**Strengths:**

The proposed method is simple and well explained, backed up with justification of why they choose the likelihood ratio strategy. The idea of using ood ideas to overcome the task-ID limitation of TIL is interesting and aligns with the continual learning community directions. The experimental results are compared with a large array of existing methods and state-of-the-art approaches.

**Weaknesses:**

The proposed method is for the most part an extension of existing previous work, which requires a replay buffer, pretrained models and the need for a forward pass for each task learned. Therefore, the advantage of not needing the task label at inference is not well contrasted with the limitations (mostly mentioned at the end of the appendix only). I would expect further discussion and justification about how these benefits and limitations balance in the main part of the manuscript.

**Questions:**

It is mentioned that HAT prevents CF, but it actually only mitigates it. It is discussed later in the appendix that a very large sigmoid is used in order to force an almost binary mask to promote more of that CF mitigation. However, how relevant is that the masks are binary and that the sigmoid is close to a step function? Would then a method that guarantees no forgetting such as PNN [Rusu et al. 2016], PackNet [Mallya et al. 2018] or Ternary Masks [Masana et al. 2020] be more suitable for the proposed strategy? How do you deal with HAT running out of capacity when the sequence gets longer?

In Table 1, which of these results are using the task label at inference time? For example, HAT needs the task label. So are the results of HAT comparable here with the other methods? Or is HAT having a forward pass with each task label and then using some heuristic to pick the class?

For the experiments on running time, in Table 9 of the appendix it is only shown the running times for the 4 methods that have the same base strategy. How do those compare with all the other methods, because I would assume that for large sequences of tasks, it might become quite a limiting factor to have to forward each sample/batch T times. I would argue that is a relevant discussion to have in the main manuscript.

In the introduction it is mentioned "This means the universal set [...] includes all possible classes in the world [...], which is at least very large if not infinite in size...". Is there some paper or relevant source to back this? One of the papers that comes to mind is [Biederman, 1987], which states that there are between 10k to 30k visual object categories that we can recognize in images. And that would hint towards learning an estimate of the distribution for objects in images would not be such unfeasible (specially now with foundational models).

In conclusion, I find the idea interesting and relevant. However, the small extension from existing related work, and the lack of a better discussion of the limitations and motivation/relevance for the community could be improved.

---

> ### Author Response · Authors · 2023-11-16
>
> Dear Reviewer fLUE,
>
> we are glad to address your concerns as follows.
>
> >  The proposed method is for the most part an extension of existing previous work, which requires a replay buffer, pretrained models, and the need for a forward pass for each task learned. Therefore, the advantage of not needing the task label at inference is not well contrasted with the limitations (mostly mentioned at the end of the appendix only). I would expect further discussion and justification about how these benefits and limitations balance in the main part of the manuscript.
>
> We think there may be a slight misunderstanding here. Our method works with or without a pre-trained model. Due to space limitations, we put only the summary results without using a pre-trained model in Table 2 (the full results are given in Table 5 of Appendix B.3). Regarding the forward pass for each task, it should not hurt the running time with parallel computing (also addressed in your third question). We have included a discussion about this in the paper on page 4 (blue font above Eq.3). Additionally, “not needing the task label at inference” is a key feature and challenge of CIL.
>
> > It is mentioned that HAT prevents CF, but it actually only mitigates it. It is discussed later in the appendix that a very large sigmoid is used in order to force an almost binary mask to promote more of that CF mitigation. However, how relevant is that the masks are binary and that the sigmoid is close to a step function? Would then a method that guarantees no forgetting such as PNN [Rusu et al. 2016], PackNet [Mallya et al. 2018] or Ternary Masks [Masana et al. 2020] be more suitable for the proposed strategy? How do you deal with HAT running out of capacity when the sequence gets longer?
>
> In our experiments, we found that the sigmoid with a low temperature is enough to prevent forgetting. We examined the task-incremental learning (TIL) accuracies and saw no forgetting. The sigmoid simulates a step function with an error less than 1e-7. The methods you suggested (PNN, PackNet, and Ternary Masks) are similar to HAT and can also be applied to our method. The main reason we chose HAT is to make it easy to compare with previous works based on the same setting. We have cited and discussed these three papers on page 4 (blue font in line 3) and Appendix E (last paragraph in blue font).
>
> The issue that HAT may run out of capacity is a good question. Actually, we tried to decrease or increase the hidden sizes of the inserted adapters used by HAT on long task sequences. We found that HAT is quite efficient in utilizing the network capacity. We believe this is due to the fact that the pre-trained model already has good features and additional adaptations in the adapters make the new task require little feature learning. We agree that the capacity will be an issue with super-long task sequences, which is a problem for all fixed architecture methods. Fortunately, it is quite simple to expand the network of HAT dynamically when needed without interfering with the learned models because its masks isolate a subnetwork for each task in a shared network.
>
> > In Table 1, which of these results are using the task label at inference time? For example, HAT needs the task label. So are the results of HAT comparable here with the other methods? Or is HAT having a forward pass with each task label and then using some heuristic to pick the class?
>
> In Table 1, our method using HAT needs the task label. But since we run all task models at test time to compute the task label probability for each task, effectively we do not need any task label. Specifically, for a test instance x, a forward pass goes through the model for each task t to compute the task label probability for the task, i.e., P(t | x) and the class probability of each class $y_j^{(t)}$ in task t, i.e., $P(y_j^{(t)} | x)$. These two probabilities are used in Eq. (1) to predict the final class for x. The baselines MORE and ROW also need task label prediction.
>
> There are also some other existing systems that need task label prediction, which we have discussed in the last paragraph of the related work section. However, we did not include them as baselines as they perform poorly (much poorer than MORE and ROW), which we have explained in the last paragraph of the related work section. For example, the two recent systems iTAML and PR-Ent give an average accuracy of 57.28 and 64.69 on all our experiments (using the same pre-trained model as TPLR), respectively, but our method TPLR gives 76.21. Their detailed results can be found in the table in our response to the third comment of Reviewer aMSi.

---

> > ### Author Response · Authors · 2023-11-16
> >
> > > For the experiments on running time, in Table 9 of the appendix it is only shown the running times for the 4 methods that have the same base strategy. How do those compare with all the other methods, because I would assume that for large sequences of tasks, it might become quite a limiting factor to have to forward each sample/batch T times. I would argue that is a relevant discussion to have in the main manuscript.
> >
> > Thanks for this suggestion! Actually, since the forward passes on different task models are independent of each other, this issue can be addressed by **parallel computing**. The running time is thus not a bottleneck here. We also add this discussion on page 4 (blue font above Eq.3) in the main manuscript.
> >
> > > In the introduction it is mentioned "This means the universal set [...] includes all possible classes in the world [...], which is at least very large if not infinite in size...". Is there some paper or relevant source to back this? One of the papers that comes to mind is [Biederman, 1987], which states that there are between 10k to 30k visual object categories that we can recognize in images. And that would hint towards learning an estimate of the distribution for objects in images would not be such unfeasible (specially now with foundational models).
> >
> > For OOD detection, the universal set is defined to include all possible classes except for the classes seen in the training data [2]. OOD detection literature acknowledges the complexity and diversity of the universal set, but we are not aware of any explicit discussion about its size. For example, an article from Borealis AI [3] discussed the extreme diversity of OOD data, acknowledging the practical difficulty in collecting sufficient data for building a classifier for OOD detection..
> >
> > We think that the paper you mentioned provides an insightful idea on the universal set of OOD detection. Also, the discussion on “OOD estimation” for foundation models is very meaningful in the post LLM-era.
> >
> > > In conclusion, I find the idea interesting and relevant. However, the small extension from existing related work, and the lack of a better discussion of the limitations and motivation/relevance for the community could be improved.
> >
> > Thank you for your support. Regarding the contribution, we would like to summarize it here again. This paper first proposes a new theoretical analysis of task-id prediction in CIL, which is based on the likelihood ratio of the distribution of a task and the distribution of the other tasks. Such an analysis has never been reported before. It points out a new direction to use the saved previous data to improve CIL performance which is entirely different from the traditional replaying. Following the theory, a principled CIL method called TPLR is proposed. Comprehensive evaluations have shown strong performance compared to state-of-the-art baselines.
> >
> > We have revised the section about the limitations of our work in Appendix H, which also puts our work in the context of existing literature. The running time issue has been discussed above.
> >
> >
> >
> > [1] Masana et al. Ternary Feature Masks: zero-forgetting for task-incremental learning, CVPR 2021.
> >
> > [2] Yang et al. Generalized Out-of-Distribution Detection: A Survey.
> >
> > [3] Borealis AI. Out-of-distribution II: open-set recognition, OOD labels, and outlier detection.
> >
> > [4] Kim et al. A theoretical study on solving continual learning. NeurIPS 2022.

---

> > > ### Comment · Reviewer_fLUE · 2023-11-22
> > >
> > > Thanks to the authors for their detailed responses. They have addressed most of my concerns, although I do not agree with the simplicity of solving the amount of computation by parallel computing (PC) not being further discussed. I do understand that that part can be solved with PC, but the main point missing in the discussion is that it might not hold well when running for very large sequence of tasks, or does not acknowledge that the computational cost is still way larger than other existing CIL methods which do not require PC.
> > >
> > > I find the suggestion by reviewer SU84 of an analysis of the influence of highly similar/dissimilar classes on same/different tasks would be interesting. In that sense, the new tables provided do not seem to tackle the issue (unless I misunderstood it). However, the response by the authors to the second question (pretrained models) is very convincing and does support the effectiveness of the proposed approach. The discussions from the other two reviewers are appreciated and I have no major concerns there.
> > >
> > > In conclusion, I still believe that the proposed method is interesting and that the provided changes help improve the manuscript. I would be in favor of acceptance.

---

> ### Author Response · Authors · 2023-11-17
>
> About your following question, we would like to clarify further.
>
> > In Table 1, which of these results are using the task label at inference time? For example, HAT needs the task label. So are the results of HAT comparable here with the other methods? Or is HAT having a forward pass with each task label and then using some heuristic to pick the class?
>
> **Answer**: The original HAT needs the task label for each test instance for task-incremental learning (TIL). However, we are using HAT for class-incremental learning (CIL) and we have **no** task label for each test instance x, so we need to adapt HAT for CIL (as we explained in the footnote of page 7).
>
> Specifically, we run x on the model of every task. For example, we have 4 tasks and we have learned 4 models for the 4 tasks based on HAT: Model1, Model2, Model3, Model4. HAT’s results in Table 1 are obtained by running x on every model via a forward pass to compute the following (for example):
>
> For Model1: class c5 in task 1 gives the highest softmax value of 0.7
>
> For Model2: class c2 in task 2 gives the highest softmax value of 0.6
>
> For Model3: class c6 in task 3 gives the highest softmax value of 0.3
>
> For Model4: class c1 in task 4 gives the highest softmax value of 0.9
>
> Then the test instance x is assigned the final class of c1 in task 4 because its highest softmax value is the highest among the 4 task models. This is similar to the "some heuristic" you mentioned in your question.

---

> ### Author Response · Authors · 2023-11-23
>
> Dear Reviewer fLUE,
>
> Thank you for your insightful feedback. We greatly value your comments and wish to discuss further about the two points you raised.
>
> 1. **Discussion on Computation by Parallel Computing:** In our previous discussion, we focused primarily on the running time and did not provide a comprehensive analysis of computational costs. To address this, we have included a new section **Appendix E.2**, analyzing computation through parallel computing. Our findings reveal that our method can attain time efficiency comparable to the standard model (without T times of forward passes), albeit with a trade-off of increased runtime memory for storing task-specific features. This trade-off enables a balance between memory usage and time consumption by adjusting the **level of parallelism**. For instance, the running time memory can be reduced by forwarding the input more frequently with a lower level of parallelism.
> 2. **Influence of Similar/Dissimilar Classes on Same/Different Tasks:** We appreciate this suggestion. We have conducted a new experiment to study this. In our paper, we used the CIFAR-100 dataset, which comprises 100 classes grouped into 20 superclasses, each containing 5 similar classes (e.g., the "flowers" superclass includes "orchids", "poppies", "roses", "sunflowers", "tulips"). We conducted the CIFAR100-20T (20 tasks) experiment using different task sequences:
>
> - seq0: each task contains classes from a distinct superclass, which means that within each task the classes are similar, but across tasks, the classes are dissimilar.
>
> - seq1: each task contains classes from different superclasses, which means that within each task, the classes are dissimilar, but across tasks, there are many similar classes.
>
> - seq2: seq3 and seq4 are randomly-mixed tasks, meaning that the classes of each task are randomly selected from random superclasses.
>
> This design should mirror your concept of "highly similar/dissimilar classes on same/different tasks." As we can see from the table below (the results are the averages of 5 random runs with 5 random seeds), the performances across these task sequences are very similar. The differences are not significant.
>
> | **Last. Acc**               | **seq0** | **seq1** | **seq2** | **seq3** | **seq4** |
> | --------------------------- | -------- | -------- | -------- | -------- | -------- |
> | mean (5 seeds)               | 76.40    | 76.25    | 76.29    | 76.32    | 76.42    |
> | standard deviation (5 seeds) | 0.28     | 0.31     | 0.38     | 0.24     | 0.33     |
>
> We thank you again for your valuable comments which help us improve our paper significantly.
>
> Sincerely,
>
> Authors

---

### Meta-Review · Area_Chair_j5og · 2023-12-06

**Metareview:**

## Overview

This paper describes an approach to class-incremental learning based on ideas from the literature on out-of-distribution (OOD) data detection. The authors propose to close the gap between task- and class-incremental learning by explicitly predicting a task-id. To do this, they use replay data to compute the likelihood ratio between current and past tasks in order to select which learned mask to apply during inference. Experimental results are provided on two benchmark datasets commonly used for class-incremental learning.

## Strengths

The paper has the following strong points:

+ **Solid motivations and theoretical foundations**: Multiple reviewers appreciate the clear motivations and the theoretical foundations given in sections 3 and 4 of the manuscript.

+ **Originality of OOD approach**: Several reviewers comment on the originality of the OOD detection approach, made possibile given the unique possibility of sampling from OOD population (exemplars from past tasks) which in traditional OOD detection is not possibile.

+ **Convincing experimental results**: Most reviewers especially appreciate the experimental evaluation and how well the approach performs on the two benchmark datasets.

## Weaknesses

The strengths cited above are balanced against by the following weak points:

+ **Poor presentation and writing quality**: Multiple reviewers commented on problems with the overall complexity of organization in the manuscript and the overuse of jargon throughout, finding that these interfered with understanding of the original submission. The authors revised the manuscript to improve these points, but the manuscript could surely use another round of revision.
+ **Novelty**: The proposed approach is basically a new OOD detection approach plugged into HAT (Hard Attention to the Task). As such, the novelty is indeed limited especially in terms of impact to class-incremental learning.
+ **Potential bias in comparative evaluation**: The authors compare against a broad variety of baselines, including exemplar-free approaches. By comparing against fundamentally non-comparable approaches, this can create the impression that the approach is much better than it actually is (i.e. it is true that it outperforms 17 baselines in Table 1, but 6 of them are not exemplar-based).

The authors engaged with two of the reviewers during the discussion stage and, in addition to providing clarifications and revising the manuscript, attempted to meet the reviewers halfway on key points by providing additional experiments and including new baselines in the comparative analysis. In the end, the main outstanding issue remains the questionable novelty given that the approach is fundamentally based on HAT with a new approach to OOD detection. The AC finds the contribution significant enough for ICLR, especially given the solid theoretical foundations which were appreciated by multiple reviewers. I encourage the authors to reorganize the paper according to reviewer suggestions and -- importantly -- to rethink the presentation of baselines from the state-of-the-art in order to make very explicit which methods are directly comparable to the proposed approach.

**Justification For Why Not Higher Score:**

The paper is clearly borderline given the reviewer opinions and the interactions between reviewers and authors during the discussion period. The proposed approach has limited novelty, with the theoretical foundations of the OOD approach applied to task selection tipping the balance from Reject to Poster.

**Justification For Why Not Lower Score:**

The experimental results are convincing, and the theoretical underpinnings of the OOD detection approach make this a good Poster for a conference like ICLR.

---

### Decision · Program_Chairs · 2024-01-16

Accept (poster)